# Clinical Value of Inflammatory and Neurotrophic Biomarkers in Bipolar Disorder: A Systematic Review and Meta-Analysis

**DOI:** 10.3390/biomedicines10061368

**Published:** 2022-06-09

**Authors:** Amanda Vega-Núñez, Carlos Gómez-Sánchez-Lafuente, Fermín Mayoral-Cleries, Antonio Bordallo, Fernando Rodríguez de Fonseca, Juan Suárez, José Guzmán-Parra

**Affiliations:** 1Instituto de Investigación Biomédica de Málaga-IBIMA, Parque Tecnológico de Andalucía, Severo Ochoa 35, 29590 Málaga, Spain; amanda_avn88@hotmail.com (A.V.-N.); gomisanchezlafuente@gmail.com (C.G.-S.-L.); fermin.mayoral.sspa@juntadeandalucia.es (F.M.-C.); antonio.bordallo.sspa@juntadeandalucia.es (A.B.); fernando.rodriguez@ibima.eu (F.R.d.F.); 2Facultad de Psicología, Campus de Teatinos, Universidad of Málaga, Andalucia Tech, 29071 Málaga, Spain; 3Unidad de Gestión Clínica de Salud Mental, Hospital Regional Universitario de Málaga, Plaza Hospital Civil s/n, Planta 1, Pabellón 4, 29009 Málaga, Spain; 4Departamento de Anatomía Humana Medicina Legal e Historia de la Ciencia, Facultad de Medicina, Universidad de Málaga, Bulevar Louis Pausteur 32, 29071 Málaga, Spain

**Keywords:** depression, mania, bipolar disorder, BDNF, cytokines, neurotrophins, biomarker, clinical factors, systematic review, meta-analysis

## Abstract

Bipolar disorder (BD) is a multifactorial chronic psychiatric disease highly defined by genetic, clinical, environmental and social risk factors. The present systematic review and meta-analysis aimed to examine the relationship between inflammatory and neurotrophic factors and clinical, social and environmental factors involved in the development and the characterization of BD. Web of Science, PubMed, PsycINFO, Scopus and Science Direct were searched by two independent reviewers. The systematic review was registered in PROSPERO (CRD42020180626). A total of 51 studies with 4547 patients with a diagnosis of BD were selected for systematic review. Among them, 18 articles were included for meta-analysis. The study found some evidence of associations between BDNF and/or inflammatory factors and different stressors and functional and cognitive impairment, but limitations prevented firm conclusions. The main finding of the meta-analysis was a negative correlation between circulating levels of BDNF and depression severity score (standardized mean difference = −0.22, Confidence Interval 95% = −0.38, −0.05, *p* = 0.01). Evidence indicates that BDNF has a role in the depressive component of BD. However, the poor consistency found for other inflammatory mediators clearly indicates that highly controlled studies are needed to identity precise biomarkers of this disorder.

## 1. Introduction

Bipolar affective disorder (BD) is a relatively common but serious mental illness, with a lifetime prevalence of 0.6% for bipolar disorder type I (BD-I), 0.4% for bipolar disorder type II (BD-II), 1.4% for subthreshold bipolar disorder (SBD), and 2.4% for bipolar disorder spectrum (BDS) [1]. It is one of the leading causes of disability worldwide [2], presenting high rates of psychiatric and medical morbidity and mortality [3,4,5,6], due to the co-occurrence with medical conditions (such as cardiovascular disorders, diabetes, and obesity), and psychiatric disorders (e.g., anxiety, personality disorders, attention-deficit/hyperactivity disorder (ADHD), substance use, and suicide) [7,8,9]. Furthermore, an early age of onset is associated with a more chronic, severe, and recurrent course of the disease [10]. In terms of gender, BD-I is equally distributed in men and women, whereas the BD-II affects more women, who are characterized by a predominance of depressive polarity [11].

BD is characterized by mood instability, and is manifested by the presence of at least one episode of mania or hypomania alternating with episodes of depression [12]. In addition to affective symptoms, neuropsychological and cognitive performance deficits, mainly related to memory, attention and executive tasks, are common [13,14,15]. These dysfunctions appear to be present from the onset of the disease, and in all mood states, even in euthymic patients and during remission [16,17,18,19]. Affective symptomatology and neurocognitive impairment interfere negatively in the psychosocial functioning and quality of life of the affected person [20,21,22].

This relationship between cognitive ability and emotionality in BD has been extensively studied, and there is evidence that poor cognitive ability, in terms of deficits in executive function and inhibition, negatively interferes with emotional regulation [23]. In addition to the effect of attentional and cognitive problems, it has been studied how emotional stimuli can influence the motor inhibition response. In particular, this response seems to be deficient in BD patients in whom certain emotions, such as sadness, may hinder the inhibition response [24]. Moreover, several studies have reported an association between cognitive performance and poorer psychosocial functioning [25,26,27,28] independently of the affective state. In addition, impulsivity has been the state most closely related to poorer functioning and more suicidal tendencies [29,30,31,32,33,34].

Along the same lines, an interesting recent study on the neural mechanisms involved in gaze perception suggests that the amygdala could play a role in attentional mechanisms, in addition to its well-known role in emotional processing [35]. Another mechanism that has been studied is the tryptophan (TRP)-kynurenine (KYN) metabolic pathway, whose action in the immune system seems to be related to the development of various diseases and psychiatric disorders that involve alterations in cognition and social functioning, including BD [36]. These findings could help to understand the observed alterations in cognition and social functioning in patients with BD.

On the other hand, new studies have recently been published on the identification of potential biomarkers in BD using mass spectrometry (MS)-based techniques. A systematic review and meta-analysis of selected studies showed evidence of alterations in the apolipoproteins (APOs) group, which appear to be associated with inflammatory response and cognitive decline. However, the results of the meta-analysis were inconclusive on the possible role of APOs as potential candidates for psychiatric biomarkers [37].

Knowledge of the etiology of BD is increasing. BD is known to be a highly genetic psychiatric disorder with a heritability of about 60–80% estimated from twin studies [38,39]. In addition, family history of BD and other biological variables (such as age, sex and metabolic factors) are relevant predictors of the development of the disorder [40]. Increasing evidence shows that the interaction between genetic and environmental factors, as well as the influence of epigenetic mechanisms, may play an important role in the etiopathogenesis of the disorder [41,42,43]. According to this hypothesis, neurodevelopmental pathways could be involved in the development of BD [44].

Furthermore, research on the role of neurodevelopmental factors in the onset of cognitive deficits in BD points to its conceptualization as a neurodegenerative disease [45,46,47]. In fact, this cognitive impairment has also been observed in first-degree relatives of BD patients [48], suggesting the presence of potential endophenotypes that could function as trait markers for the disorder [49,50]. Similarly, many efforts have been made so far in the search for possible biological mechanisms underlying different neurological and psychiatric disorders, such as depression, anxiety and dementia, with which BD shares some symptomatic manifestations [51].

The nature of BD is complex and is determined by multiple causes. One of the proposed mechanisms involved in the neurobiology of the disorder is neuroinflammation [52,53,54]. Numerous studies have also found differences in the levels of trophic factors of the neurotrophin family (mainly brain-derived neurotrophic factor (BDNF)) and cytokines (such as interleukins (e.g., IL-6, IL-1β, IL-2, IL-4, IL-8, IL-10, IL-18) and tumor necrosis factor alpha (TNF-α)), as well as the influence of oxidative stress mediators in BD patients compared to healthy controls [55,56]. Supporting these findings, BDNF could play an important role in the alterations in neuroinflammation, neuroplasticity and neurogenesis for BD [57]. However, no specific biomarker has yet been identified for each phase of the disorder [58].

Regarding the identification of genetic markers for BD, a candidate gene strategy has been largely unsuccessful [59]. BDNF is one of the most investigated candidate genes by studies looking at genetic (GWAS) and epigenetic (DNA methylation and EWAS) variations associated with the risk of BD [60,61]. Evidence from longitudinal analysis of methylation profiles support a role of inflammatory factors (e.g., BDNF, IL-1β) in the pathogenesis of BD [60]. However, potential confounders (underpowered sample sizes, inadequate testing correction, undetected genetic mismatch, global methylation or inconclusive results compared to other psychiatric disorders) may explain the lack of consistent findings and the difficulties in establishing BD susceptibility genes.

Furthermore, psychological and social factors must be taken into account. Indeed, a substantial body of studies indicates that there is a relationship between BD and exposure to clinical, environmental, and social risk factors [62,63,64,65] that precede and could predict the onset of BD. Exposure to environmental risk factors during prenatal and perinatal periods [66,67], childhood maltreatment or abuse [68,69,70], stressful life events [71,72,73], and alcoholism or substance abuse [74,75] are some of the factors that can be phenomenologically related to the development of BD. Interestingly, such early life stressful events could also promote a neuroinflammatory state that would eventually contribute to the onset of BD [76].

Although the existing literature demonstrates relationships between inflammatory mediators and bipolar disorder as compared to healthy controls, a systematic review of a reliable biomarker signature for bipolar disorder that could help to distinguish between different subtypes, phases (depression, mania and euthymia) and disease severities, has not been accurately performed. The aim of this study was to conduct a systematic review and meta-analysis to identify cytokines, inflammatory and neurotrophic mediators involved in the development of BD. The effort to structure and expand knowledge in this area of study has important implications for improving the knowledge of the underlying biological basis of the disorder. Thus, the clinical value of inflammatory biomarkers has to be demonstrated by highly controlled studies that follow a precise phenotyping (type, phase, severity, and comorbidities) of BD patients. The identification of accurate biomarkers could improve the objectivity of BD diagnosis and the validity of treatment in clinical trials.

## 2. Materials and Methods

### 2.1. Search Strategy

This systematic review and meta-analysis were carried out following the recommendations of the Preferred Reporting Items for Systematic Reviews and Meta-Analyses (PRISMA) statement [77]. The protocol was registered in the International Prospective Register of Systematic Reviews (PROSPERO CRD42020180626). An initial review was conducted to identify biomarkers involved in bipolar disorder to generate a list of search terms.

Bibliographic searches were performed in the electronic databases Web of Science, PubMed, PsycINFO, Scopus and Sciencedirect from their inception to May 2020. The database searches were complemented with manual screening of the reference list of the included studies.

Two independent searches were achieved on each database (conducted by AV-N and CG-S-L). Detailed information about the search strategy can be found in the Appendix A.

The inclusion criteria applied were the following: (a) type of study: clinical trial, cohort and case-control studies; (b) age: adults subjects (patients over 18 years old); (c) diagnosis: any bipolar disorder (BD-I, BD-II, Cyclothymic Disorder and not otherwise specified BD (NOS-BD)) in any of its phases (mania, depression, mixed state or euthymia) diagnosed according to international classification systems, including clinical interviews that applied the criteria of the DSM (Diagnostic and Statistical Manual of Mental Disorders) or the ICD (International Classification of Diseases); (d) studies that include exposure to clinical, social or environmental factors; (e) studies investigating some type of inflammatory factors (biomarkers indicative of a biological state in the person, in this case the degree of brain inflammation), neurotrophic factors (or neurotrophins, a family of proteins that promote the survival, growth and differentiation of neurons; among them are nerve growth factor (NGF), the insulin-like growth factor (IGF-1), the BDNF and neurotrophin-3 (NT-3)), and cytokines (a set of IL that regulate the immune and inflammatory response) in patients with bipolar disorder; (f) language: English or Spanish only.

The following exclusion criteria were applied: (a) narrative reviews, qualitative studies and case studies; (b) children and adolescents (under 18 years old); (c) subjects diagnosed with another mood disorder (e.g., major depressive disorder, dysthymic disorder, unspecified depressive disorder); (d) subjects diagnosed with any other psychiatric disorder; (e) subjects with a diagnosis of neurological disorder or genetic syndromes; (f) studies that not include exposure to clinical, social or environmental factors related to BD; (g) studies in which the only exposure factor was a pharmacological treatment or psychological intervention; (h) studies that have not investigated neurotrophic factors and inflammatory cytokines; (i) studies with duplicated or unoriginal data.

### 2.2. Study Selection

Title and abstract were screened independently by two reviewers (AV-N and CG-S-L). If the presence of the biomarkers, risk factors or study design could not be ascertained from title and abstract screening, full texts of publications selected were then reviewed by the same investigators to determine final eligibility. Team meetings were held to discuss and resolve any discrepancies and reach a consensus on all inclusion decisions with a third reviewer (JG-P). If the data needed to include the study in the meta-analysis were not available, authors were contacted up to two times (two weeks apart). If no response was received, these studies were not included in the meta-analysis but were discussed narratively.

### 2.3. Quantitative Analysis

A meta-analysis was performed for all factors reported on by at least three studies. Mean and standard deviation for each biomarker, separated into mood phase or subtype of bipolar, were entered into an electronic database and were analyzed with a quantitative meta-analytical approach using R 3.6.3 and Rstudio. The mean differences and 95% confidence intervals between cases and controls were calculated and displayed in forest plots. Correlations between severity scales and circulating levels of biomarkers were transformed to Fisher’s Z-score to obtain accurate weight for each study. Montgomery-Asberg Depression Rating Scale (MADRS) scores were translated to Hamilton Depression Rating Scale (HDRS) scores using the equipercentile linking method developed by Leucht and colleagues [78]. The inverse variance method and Random-effects models by Hartung-Knapp-Sidik-Jonkman were used. This method produces more robust estimates and is preferred when the number of studies is small and when there is substantial heterogeneity, because DerSimonian-Laird is prone to produce false positives under this condition [79,80]. Heterogeneity was determined by means of Q statistic tests and the I2 index. Publication bias was evaluated by use of funnel plots in each meta-analysis including ten or more studies, as is recommended for interpretation [81]. Studies included in the meta-analysis were appraised using the Risk of Bias in a Non-randomised Studies (ROBINS-I) tool [82].

## 3. Results

A total of 4396 studies were retrieved from the databases. Finally, 51 articles (4 of which had repeated samples with other biomarkers) that analyzed 33 inflammatory mediators were eligible for review and 18 were included in the meta-analysis (Appendix A). More detailed information is shown in the PRISMA flow chart (Figure 1). Risk of bias assessment of the studies included in the meta-analysis is summarized in a traffic light plot (Appendix A) and the general risk of bias of all studies included in the meta-analysis is shown in a summary plot (Appendix A). The overall risk of bias of the studies included was between moderate and serious. Among the included studies, the most commonly reported outcomes were BDNF and mood state, BDNF and bipolar subtype, correlation between BDNF and severity of mania or depression, TNF-α, and mood state, correlation between TNF-α and severity of mania, correlation between IL-6 and severity of mania or depression. We selected these outcomes and performed 9 meta-analyses.

The characteristics and main results of the 51 studies included in the systematic review are summarized in Table 1. The studies were conducted across several countries. The design of the studies was mostly cross-sectional, although clinical trials, post-hoc analysis of a clinical trial, longitudinal studies, retrospective and prospective studies, a historical cohort study, and a naturalistic cohort study were also included. More information on the studies can be found in the Appendix A.

The studies comprised a total of 4547 patients with diagnoses of BD-I, BD-II and NOS-BD. Most studies used the Structured Clinical Interview for DSM-IV Axis I Disorders (SCID-I) or the Mini-International Neuropsychiatric Interview (MINI) to assess bipolar diagnosis, but the Structured interview of the Modified Schedule of Affective Disorder and Schizophrenia-Life Time (SADS-L) and the Schedules for Clinical Assessment in Neuropsychiatry (SCAN) were also used. Symptom severity was assessed with the Young Mania Rating Scale (YMRS) for mania or hypomania and the HDRS or the MADRS for depression. The patients were examined at different phases of the disease (euthymia, mania, depression, mixed) and in different stages of the disorder (early stage and late stage). The biomarkers evaluated were BDNF, IL-2, IL-4, IL-6, IL-8, IL-10, IL-18, IL-33, IL-1α, IL-1β, IL-1Ra, sIL-2R, sIL-6R, IL-18BP, TNF-α, IFN-γ, IGF-1, β-NGF, GDNF, VEGF, TGF-β1, TNFR1, TNFR2, TNF-R2, sTNFR60, sTNFR80, NT-3, NT-4/5, sICAM-1, sVCAM-1, sFlt-1, sSt2 and sCD40L (see Appendix A for abbreviations).

### 3.1. Stratifying Value of Biomarkers: Analysis of Clinical Features of BD

#### 3.1.1. Subtypes of the Disorder

A total of 11 articles (1, 17, 21, 25, 31, 32, 39a, 39b, 40, 44a, 44b in Table 1), included in the present systematic review, evaluated the association between BD subtypes I and II and selected biomarkers, 7 studied BDNF and 9 measured inflammatory biomarkers. Only one article found differences in BDNF levels between BD-I and BD-II subtypes (40 in Table 1), in which patients with the BD-I subtype had significantly lower BDNF levels than patients with the BD-II subtype. Only one article found differences in the circulating levels of inflammatory mediators (sIL-2R and sTNF-R1), which were higher in BD-I patients than in BD-II patients (1 in Table 1). There were two more articles that compared the subtypes of BD with the SBD group (subthreshold bipolar disorder: more than 2-days but less than 4-days duration of hypomania) and found differences in the levels of inflammatory mediators (IL-1ß, IL-6, IL-8) and BDNF (45, 46 in Table 1).

In the present study, we performed a meta-analysis of BNDF between BD subtypes in three selected studies (21, 40, 45 in Table 1). These studies highlighted lower plasma BDNF levels in BDI (40 in Table 1) and SBD (45 in Table 1) patients, and in BD women with higher MADRS scores (21 in Table 1). In the meta-analysis, we found that circulating BDNF levels did not differ between BD subtypes I and II (standardized mean difference [SMD] = 0.24, 95% CI = −1.19, 0.71, *p* = 0.39, number of studies [k] = 3, number of participants [n] = 645) (Figure 2).

#### 3.1.2. Phases of the Disorder

A total of 16 articles (1, 2, 3, 5, 8, 18, 19, 32, 33a, 33b, 34, 39a, 42, 44a, 44b, 47 in Table 1), included in the present systematic review, studied the association between biomarkers and different mood phases in BD patients, 8 of them assessed inflammatory factors, 5 assessed BDNF and other trophic factors, and 3 both biomarkers. Among the studies in which inflammatory mediators were measured, 5 of them found significant differences with respect to episode mood between depressive and (hypo)manic state (IL-4, IL-6, IL-18, IL-1β, sIL-2R, sTNF-R1, sTNF-R80) (1, 34, 33b, 39a in Table 1), 2 studies compared depressive and euthymic patients (sIL-2R, sTNF-R1, sTNF-R80) (1, 39a in Table 1), 1 study compared (hypo)manic and euthymic patients (IL-6, IL-18) (33b in Table 1), and one study compared patients with mood symptoms (depressive and (hypo)manic) and euthymic (TNF-α) (44a in Table 1). Among the studies in which BDNF and other trophic factors were evaluated, only two of them found differences in circulating BDNF levels between manic/depressed and euthymic BD patients (42 in Table 1) and between manic and depressive BD patients (47 in Table 1).

In addition, we performed a meta-analysis of BNDF between phases of the disorder from the data of five selected studies (2, 18, 32, 33a, 42 in Table 1). Most of these studies reported no differences in plasma BDNF levels between affective states (2, 18, 32, 33a in Table 1), and only one showed lower BDNF levels in depressive and manic patients compared to euthymic patients (42 in Table 1). In our meta-analysis of trophic factors between phases of the disorder, we found that circulating BDNF levels showed no difference in the depressive state compared to euthymia (SMD = −0.57, 95% CI = −3.45, 2.31, *p* = 0.48, k = 3, n = 313) (Figure 3a), nor in mania compared to euthymia (SMD = −0.26, 95% CI = −0.87, 0.36, *p* = 0.31, k = 5, n = 452) (Figure 3b). We also performed a meta-analysis of TNF-α between BD phases from three selected studies (18, 34, 44a in Table 1). Only one of them (44a in Table 1) reported increases in plasma TNF-α levels of patients with moods symptoms compared to euthymic patients. In the meta-analysis, circulating TNF-α levels show no difference in depressive patients compared to mania patients (SMD = −0.02, 95% CI = −0.69, 0.64, *p* = 0.91, k = 3, n = 167) (Figure 3c).

#### 3.1.3. Severity of Mood Symptoms

A total of 30 articles (1, 2, 3, 5, 7, 8, 11, 12, 13, 17, 19, 21, 22, 23, 25, 26, 27, 28, 30b, 33a, 33b, 35, 38, 39a, 40, 42, 43, 44a, 44b, 46 in Table 1) studied the relationship between symptom severity and the biomarkers included in the present systematic review. Among them, 24 reports assessed the severity of manic symptoms and 26 reports assessed the severity of depressive symptoms. In terms of biomarkers, 14 studies measured inflammatory mediators, 14 measured BDNF and/or other trophic factors, and 2 measured both biomarkers. Seven studies reported an association between circulating levels (11, 26, 27, 42 in Table 1) of BDNF and another trophic factor (NGF) (3) and severity of manic symptoms, as well as between trophic factors (BDNF, GDNF) and the severity of depressive symptoms (12, 21, 42 in Table 1). On the other hand, several studies reported an association between inflammatory mediators (sVCAM-1, TNF-α, IL-2, IL-6, sIL-6R, sTNF-R1, TGF-ß1) and severity of manic symptoms in 6 studies (1, 13, 8, 23, 35, 46 in Table 1) (inflammatory mediator with growth properties), as well as between inflammatory mediators (sTNFR60, sTNFR80, sVCAM-1, IL-6, IL-1ß, TNF-α, sIL-6R, sIL-2R, sTNF-R1, TNF-R1, TGF-ß1) and severity of depressive symptoms in 9 studies (1, 8, 13, 19, 23, 28, 35, 44a, 46 in Table 1). Another study found higher GDNF levels in manic and depressed BD patients when compared to euthymic patients (38 in Table 1).

Among the above studies, 5 of them evaluated the global disease severity (12, 17, 19, 35, 42 in Table 1) in which inflammatory mediators, BDNF and GDNF were evaluated, and 4 of them found a connection between plasma biomarker levels (sVCAM-1, TNF-α, BDNF) and disease severity (12, 19, 35, 42 in Table 1).

In the present study, we performed a meta-analysis between BNDF and severity of mood symptoms from the data of seven selected studies (11, 12, 21, 22, 26, 40, 42 in Table 1). These studies reported contradictory results; plasma BDNF levels positively correlated with YMRS scores among BD manic patients (11 in Table 1), and negatively correlated with YMRS, HDRS and/or CGI scores in BD patients (12, 42 in Table 1), with MADRS scores in BD women patients (21 in Table 1) and with HDRS and CGI scores among the BD depressed group (12 in Table 1), and no correlation was observed between BDNF and YMRS scores (22, 40 in Table 1). Similarly, no correlation between TNF-α and mania severity score (CGI, MADRS, HDRS, YMRS and/or PANSS) were reported in the three studies selected. Meta-analysis indicated that circulating BDNF levels showed a significant negative correlation with the severity of depression (SMD = −0.24, 95% CI = −0.37, −0.10, *p* = 0.01, k = 5, n = 271) (Figure 4a).

We also performed a meta-analysis between IL-6 and severity of mood symptoms from five selected studies (13, 30, 35, 43, 44b in Table 1). Only one of these studies reported a positive correlation between plasma IL-6 levels and severity of depressive symptoms in BD patients (13 in Table 1). The remaining studies reported that plasma IL-6 levels did not correlate with MADRS, YMRS, CGI, HDRS and/or PANSS scores in BD patients (30b, 35, 43, 44b in Table 1). In the meta-analysis, no correlation between depression severity and IL-6 was found (SMD = 0.10, 95% CI = −0.12, 0.31, *p* = 0.38, k = 4, n = 181) (Figure 4a).

Severity of mania did not correlate with BDNF (SMD = −0.25, 95% CI = −0.60, 0.18, *p* = 0.25, k = 4, n = 372), IL-6 (SMD = 0.08, 95% CI = −0.14, 0.30, *p* = 0.48, k = 4, n = 202), nor TNF-α (SMD = 0.04, 95% CI = −0.22, 0.30, *p* = 0.76, k = 3, n = 180) (Figure 4b).

#### 3.1.4. Factors Related to Clinical Staging of the Disorder

The length of illness was assessed in 17 (1, 2, 3, 5, 8, 9, 17, 19, 25, 31, 32, 33a, 39a, 40, 44a, 44b, 46 in Table 1) of the articles included in the systematic review, 8 of them evaluated inflammatory mediators, 6 reports analyzed BDNF and other trophic factors and 3 of them evaluated both biomarkers. Length of illness was associated with inflammatory mediators (IL-6, sIL-6R, sIL-2R, sTNF-R1, TNF-α) in 4 studies (1, 32, 44b, 46 in Table 1), and with BDNF and other trophic factors (NT3, NT4/5, NGF) in 3 other studies (3, 25, 33a in Table 1). In one study, although correlation between BDNF and overall disease duration was not observed (2 in Table 1), significant differences were found when the groups were segregated into more than 10 years and less than 10 years of disease duration.

Age at onset was assessed in five articles (31,35,39a,42,44a in Table 1). Two of them found an association between age of onset and inflammatory (sVCAM-1) and trophic (GDNF) factors (35, 42 in Table 1). No relationship was found between biomarkers and the number of hospitalizations in any study (3, 44a, 44b in Table 1). No interaction was also found between BDNF levels and chronicity and time [7]. Levels of sVCAM-1 correlated negatively with the bipolarity index (BPx) in the acute and remission phases (35 in Table 1). Another inflammatory mediator (TNF-α) was associated with the duration of untreated BD only in one study (35 in Table 1). Among the 3 studies (15, 19, 41 in Table 1), in which they classify BD patients into early or late stage, all measured inflammatory mediators, 2 of them evaluated BDNF and the other analyzed additional trophic factors. These three studies found significant differences in circulating levels of inflammatory mediators (IL-6, IL-10, IL-1RA, TNF-α) when comparing early- and late-stage patient groups, but not when analyzing circulating levels of trophic factors.

#### 3.1.5. Number of Previous Episodes, Duration of Episodes and Polarity

Regarding the number of previous episodes, the results are heterogeneous. Some studies evaluated the number of lifetime mood episodes, but no association with biomarkers was found (39a, 44b in Table 1). Only one article associated the number of depressive episodes with inflammatory factors (IL-6) (37 in Table 1), but not with BDNF (31 in Table 1) or with the presence of previous episodes of depression (4 in Table 1). In one study, a longer duration of mood episodes (mixed, manic, depressive, euthymic) was associated with inflammatory mediators (IL-6, IL-18) in comparison with episodes of less than one-week duration (33b in Table 1). However, no association was found between mood duration and BDNF or other trophic factors (GDNF) (42 in Table 1). Another study found an association between the duration of current mood state and inflammatory and trophic factors (TNF-α, BDNF, VEGF and sFlt-1) (44a in Table 1), but this result was not confirmed (44b in Table 1). The duration of depressive episodes was associated with inflammatory mediators (TNF-α) in only one study (35 in Table 1).

No associations were found between any biomarker and rapid cycling bipolar patients or atypical depression (20, 39a in Table 1). Furthermore, no differences in circulating levels of inflammatory mediators (sTNF-R1, sIL-6R) were found between patients with subsyndromal BD and without subsyndromal symptoms (9 in Table 1). Similarly, no differences in mean levels of sICAM-1, sVCAM-1, IL-6 and TNF-α were observed when comparing patients in an acute phase and patients in a remission phase of the bipolar disorder (35 in Table 1). Regarding polarity, BD patients with manic polarity had significantly lower levels of sVCAM-1 than BD patients with depressive polarity (35 in Table 1). No differences were found in terms of acute episode polarity and levels of sICAM-1, IL-6 or TNF-α (35 in Table 1).

#### 3.1.6. Cognition and Functionality

Four articles evaluated general cognitive abilities in BD patients, including cognitive and executive functioning, 1 with inflammatory mediators, 1 with BDNF and 2 with both types of factors. Among these reports, significant associations were found between cognitive abilities and inflammatory mediators (IL-1RA, sCD4DL, TNF-α) (4, 16 in Table 1) and BDNF (24, 32 in Table 1). On the other hand, two studies evaluated the degree of functionality. In one, functional impairment was associated with inflammatory mediators (IL-6, IL-10) (30b in Table 1), and in the other was reported a significant interaction between psychosocial functioning and time on BDNF levels (7 in Table 1). In one study, BDNF was not associated with quality of life (24 in Table 1). Finally, a positive correlation between BDNF levels and Global Assessment of Functioning (GAF) scores was found in one study (42 in Table 1), but not when the trophic factor GDNF was analyzed.

#### 3.1.7. Other Clinical Parameters and Comorbidity

Nine articles evaluated age in BD patients, which was associated with inflammatory mediators (IL-6, IL-8, IL-18, IL-18BP, TNF-α, sTNFR1, sTNFR2) in 3 of these studies (10, 17, 44b in Table 1). Only two of these articles included sex as a study variable, but it was not correlated with any biomarker (17,28 in Table 1).

Only one article found an association between the number of previous psychotic episodes and inflammatory mediators (sVCAM-1) (35), and others did not find any association between biomarkers and lifetime features of psychosis (4, 17, 44a, 44b in Table 1). The presence of psychotic symptoms was not associated with inflammatory mediators or BDNF in any study (19, 25, 39a, 44a, 44b in Table 1).

Psychiatric comorbidities were generally not associated with inflammatory mediators or BDNF (2, 3, 4 in Table 1), with the exception of one study (46 in Table 1) that found a relationship with TGF-ß1. Melancholia was associated with inflammatory mediators (sTNFR80, IL-1α) in the two selected studied (39a, 39b in Table 1). Comorbid anxiety symptoms were associated with inflammatory mediators (IL-6, IL-10, TNF-α) in one study (14 in Table 1), but another study found no association between comorbid anxiety syndrome and inflammatory mediators (17 in Table 1). Post-traumatic stress disorder was not associated with the biomarkers assessed (20). The history of suicide attempts was not associated with BDNF (20 in Table 1), nor was suicidal ideation with inflammatory factors (39a in Table 1). ADHD was not associated with the biomarkers evaluated (17 in Table 1). Tobacco use, substance abuse or dependence, and alcohol abuse or dependence, were not associated with any biomarker in the studies evaluated (3, 4, 17, 20, 25, 28 in Table 1).

Childhood abuse and neglect and childhood trauma were associated with BDNF in the two studies that assessed this biomarker (6, 20 in Table 1), but in two other studies inflammatory factors were not related to these variables (28, 36 in Table 1). No association was found between resilience and BDNF (29 in Table 1). An inflammatory factor (sVCAM-1) was associated with family history of BD in one study (35 in Table 1).

Chronotype was associated with an inflammatory biomarker (IL-6) in one study (30a in Table 1). As for sleep disorders, they were associated with the inflammatory factor IL-6 in one study (13 in Table 1). Dietary intake was associated with inflammatory factors (sTNFR1, sTNFR2) in one study (10 in Table 1), and weight cycling was associated with inflammatory factors (IL-6) in another (37 in Table 1). Two studies included BMI, but no correlations with inflammatory factors were found (17, 28 in Table 1).

## 4. Discussion

This systematic review and a series of exploratory meta-analyses represent an effort to synthesize reliable evidence on the association of inflammatory and trophic factors with subtypes, phases and severity of the disorder, in addition to factors related to clinical staying, episodes, duration, polarity, among other clinical parameters and comorbidities related to BD. We were able to include 51 studies, comprising 4547 patients with BD. The main finding of this systematic review and meta-analysis is the negative correlation found between BDNF levels and severity of depressive symptoms in BD patients. This evidence suggests that BDNF may be an eventual biomarker for BD.

This result is consistent with one of the most replicated findings in the literature, the decreased peripheral levels of BDNF and its association with depressive symptoms [83,84]. Besides, a positive correlation has recently been found between severity of manic symptoms and BDNF levels in BD patients [85]. Moreover, other studies find lower BDNF levels in BD patients in manic and depressive phases compared to controls [86], but these differences were not significant across affective states in general [87]. Although evidence suggests that BDNF has an important role in the psychopathology and progression of BD [88], there are inconsistent findings due to a number of confounding factors in uncontrolled studies that highlight the need for specificity as a biomarker in the diagnosis of BD. BDNF plays an increasingly important role as a biomarker not only in affective disorders, but also in cognitive impairment, such as that associated with alcohol abuse. It will be important to identify whether the decrease in BDNF observed in BD could be associated with present or ulterior cognitive impairment in BD [89].

The present review and meta-analysis do not show consistent associations between the inflammatory biomarkers studied and the main psychiatric and clinical variables of the disorder. Persistent pathogens likely support a role of an altered immune system in the etiology of mood disorders [90]. In particular cases such as TNF-α, the previously proposed association with BD is not supported by the present meta-analysis, and this finding is supported by recent studies with monoclonal antibodies against this cytokine that showed no effects in the prefrontal neurochemistry of patients with BD [91]. In a systematic review and exploratory meta-analyses in which levels of neurotrophic, inflammatory and oxidative stress biomarkers were analyzed in combination as a function of an affective state (euthymia, mania, depression), no biomarkers were found that could be individually discriminative of each mood phase, with the combination of hsCRP/IL-6, sTNFR1 with BDNF/TNF-α being significant [58]. However, later studies suggest that elevated levels of CRP and TNF-α in manic, depressive, and mixed episodes would point to these substances as biomarkers of mood episode, whereas elevated IL-6 levels during euthymic could refer to a euthymic role as a trait biomarker in BD [92].

Overall, heterogeneity and inconsistency in previous studies are supporting the search for new potential targets with a clinical value in BD. Several additional inflammatory factors, such as specific chemokines (MCP-1, fractalkine), glial factors (GFAP, S100B), or lipid-derived mediators such as endocannabinoids, and neurodegeneration-related molecules such as neurofilament light chain, neurogranin and β-amyloid peptide isoforms (Aβ42, Aβ40, Aβ38) and their fragments could be promising biomarkers in patients with BD in relation to prospective clinical outcomes, such as those of neuronal injury, cognitive deficits and risk of dementia [93]. Recent studies have also evaluated non-coding RNA expression, sexual hormones, oxidative stress (uric acid, bilirubin, albumin) and metabolic biomarkers (metabolic syndrome, overweight/obesity, thyroid and liver function) in predicting BD and suicidality in individuals with bipolar disorder [94,95,96,97,98]. In addition, big data of cerebral blood flow and structural and functional connectivity using magnetic resonance imaging are becoming useful tools for actual practice in psychiatry [99,100]. By gathering all this information and using multivariable logistic regression, it could be possible to construct a model to predict BD.

Other results, such as the correlation between the biomarkers and bipolar subtypes, stage of disease, severity of manic symptoms and global severity of illness, were heterogeneous and should be analyzed with caution. Some studies found an association between biomarkers and other clinical variables included in the study, such as stressful life events, childhood trauma, cognitive abilities and functionality, but further studies are needed to confirm the relationship between inflammatory and neurotrophic factors and these variables. In general, more studies are necessary to understand precisely the neurobiology of BD and their interactions with stress and clinical factors. This improvement will presumably translate to more sophisticated treatments.

Taken together, although biomarkers and clinical, social and environmental factors have been shown to play an important role in understanding the etiopathology and development of bipolar disorder, these variables are not clinically replicable [101]. The clinical value of BDNF and other inflammatory biomarkers has yet to be demonstrated by highly controlled studies that follow a precise phenotyping (type, phase, severity, and comorbidities) of BD patients. Moreover, other promising biomarkers related to mitochondrial dysfunction, oxidative stress, metalloproteinases, HPA axis function, autoimmunity and urinary metabolites, among others, need to be studied [102,103].

### Strengths and Limitations

This comprehensive systematic review includes a series of exploratory meta-analyses of neurotrophic and inflammatory factors in bipolar disorder. The results for each biomarker have been separated by clinical and social factors, attempting to provide information on the physiological changes that occur with the illness, and reducing the risk of type 2 error. The use of random effects models adds further validity to the results.

However, this review has several limitations and the results should be interpreted with caution. First, most of the studies were cross-sectional, and changes in trophic/inflammatory biomarkers throughout the development of BD in an individual patient cannot be described. Most of the studies reviewed were observational and at significant risk of bias and confounding, so causality cannot be established. Individual characteristics varied greatly between studies. Many studies included small samples and the aims between them were different. Some factors influencing biomarker levels, such as body mass index, physical activity, drug use, or blood pressure, were not addressed in many studies [104,105]. Some studies measured several cytokines, but the results were not reported due to a lack of sensitivity in assays that detected valid levels. Dilution and storage of blood samples and measurement factors in plasma or serum may also influence the variability and reproducibility of biomarker levels [106,107,108]. The influence of medication is not analyzed in most studies, while others include drug-naïve patients. The inclusion of patients followed different criteria for defining bipolar disorder (i.e., 2 days of hypomania vs. 4 days) and the variation in YMRS and depression scale scores for defining mood phase probably contributed to the heterogeneity of the results. Several clinical variables were assessed from clinical interviews only, which may compromise the reliability and validity of the results. Because of the need to separate results by clinical and social factors, a significant number of meta-analyses were performed, which increases the risk of type 1 error. The decision to perform the meta-analysis with only three studies is arbitrary, but we have taken this decision to increase our statistical power by reducing the standard error of the weighted average effect size. The purpose of our review was exploratory, so multiple testing was not corrected for, and the results should be considered with caution. In addition, it is likely that some meta-analyses were underpowered as a result of a small sample size and the small number of studies included. To determine the validity and precision of a potential biomarker, future methodologically consistent and repeatable studies are needed to allow homogeneity and comparability of results.

## 5. Conclusions

The present study shed light on the evidence supporting inflammatory and neurotrophic factors as reliable biomarkers that can characterize clinical features for BD. At the moment, the mixed and contradictory evidence regarding the association of inflammatory and neurotrophic factors with clinical features suggest that highly controlled studies with wide sample sizes need to be conducted. Probably, heterogeneous and small samples, selection bias, different methodologies and bias in the measurement of outcomes, lack of control regarding confounders and bias in the selection of the reported results could explain the mixed evidence found in this review. The meta-analysis clearly indicates that BDNF has a role in the depressive component of BD and more studies need to be conducted to establish the clinical value of this biomarker. The systematic review suggests a possible association between stressors, functional and cognitive impairment, and BDNF and/or inflammatory factors but the small number of studies do not permit to establish solid conclusions. Moreover, the poor consistency of promising inflammatory mediators such as IL-6 and TNF-α indicates the need for homogeneous studies that help identify additional biomarkers of this disorder.

## Figures and Tables

**Figure 1 biomedicines-10-01368-f001:**
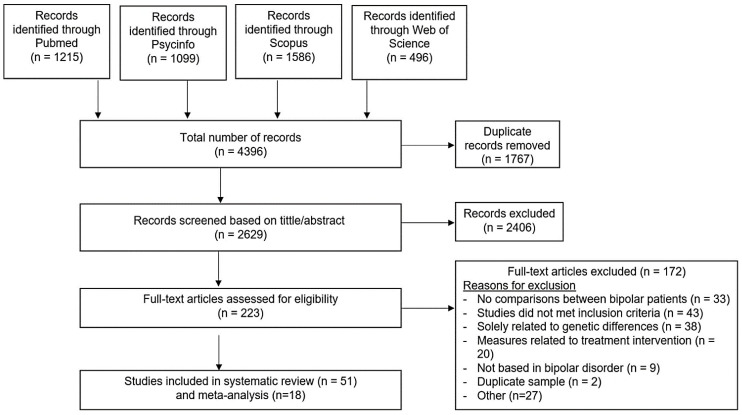
PRISMA (Preferred Reporting Items for Systematic reviews and Meta-analyses) diagram. Following exclusion criteria, a total of 51 studies were selected for systematic review. Among them, 18 articles were included for meta-analysis.

**Figure 2 biomedicines-10-01368-f002:**
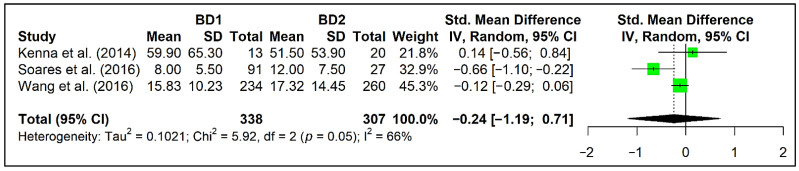
Forest plot of the meta-analysis of BDNF in patients with BD1 subtype compared to those with BD2 subtype in three selected studies (21, 40, 45 in Table 1).

**Figure 3 biomedicines-10-01368-f003:**
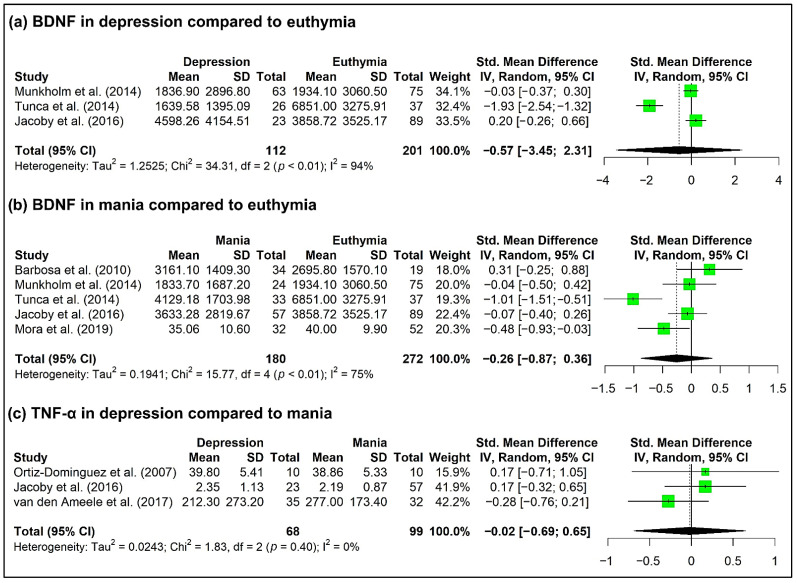
Forest plot of the meta-analysis of BDNF in (**a**) depression compared to euthymia from the data of three selected studies (18, 33a, 42 in Table 1), (**b**) mania compared to euthymia (**b**) from the data of five selected studies (2, 18, 32, 33a, 42 in Table 1), and (**c**) TNF-α in depression compared to mania from the data of three selected studies (18, 34, 44a in Table 1).

**Figure 4 biomedicines-10-01368-f004:**
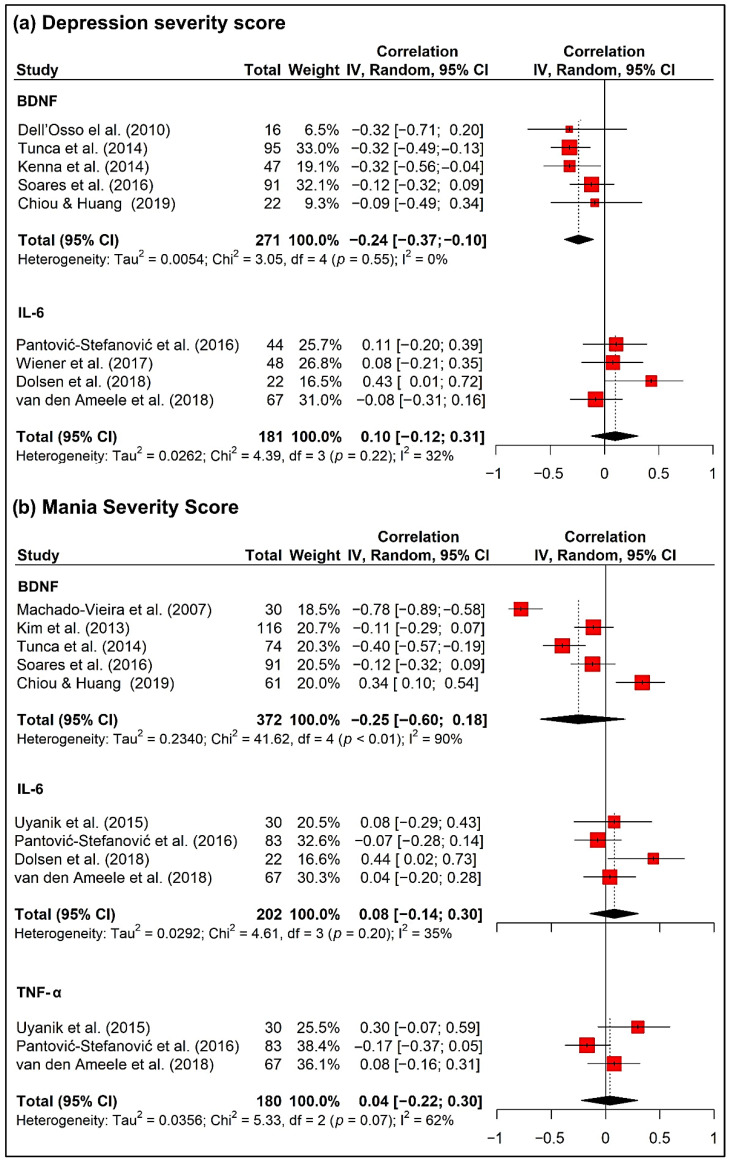
Forest plot of correlations between inflammatory factors (BDNF, IL-6 and TNF-α) and (**a**) depression severity score (HDRS or MADRS) from the data of selected studies (11, 12, 13, 21, 30b, 35, 40, 42, 44a in Table 1) and (**b**) mania severity score (YMRS) from the data of selected studies (11, 13, 22, 26, 35, 40, 42, 43, 44a in Table 1).

**Table 1 biomedicines-10-01368-t001:** Summary of the 51 studies analyzing 33 biomarkers in 4547 patients with a diagnosis of BD included in the systematic review and meta-analysis.

ID	First Author (Year)	BD Sample	BD Age (Range)	BD Sex (M/F)	Exposure Factor	Measure Instrument	Biomarkers Evaluated	Main Findings
1.	Bai et al. (2014)	130 BD patients (77 BD-I, 53 BD-II) (75 euthymic, 14 (hypo)manic state, 41 depressive state)	44.55 ± 11.08 (47.0 ± 11.7 BD-I, 41.0 ± 10.2 BD-II) (44.9 ± 12.4 euthymic, 46.1 ± 9.3 (hypo)manic, 43.4 ± 10.2 depressive)	31.55%/68.44% (33.8%/66.2% BD-I, 28.3%/71.7% BD-II) (33.3%/66.7% euthymic, 21.4%/78.6% (hypo)manic, 33.3%/66.7% depressive)	Severity of manic symptoms Severity of depressive symptoms Subtype, phase of the disorder and other clinical features	YMRS MADRS MINI	sIL-2R sIL-6R sTNF-R1	BB-I patients had higher levels of sIL-2R and sTNF-R1 than BD-II patients. Lower levels of sIL-2R and sTNF-R1 in depressive state than in a manic/hypomanic and euthymic state. Levels of sIL-6R and sTNF-R1 correlated positively with YMRS scores. Levels of sIL-6R, sIL-2R and sTNF-R1 correlated negatively with MADRS scores. Levels of sIL-6R, sIL-2R, sTNF-R1 correlated positively with length of illness.
2.	Barbosa et al. (2010)	53 BD-I patients (34 mania, 19 euthymia)	47.77 ± 13.01 (49.6 ± 14.2 mania, 44.5 ± 10.9 euthymia)	39.59%/60.40% (38.2%/61.8% mania, 42.1%/57.9% euthymia)	Severity of manic symptoms Severity of depressive symptoms Phase of the disorder, comorbidity and other clinical features	YMRS HDRS-17 MINI-plus	BDNF	No differences in BDNF levels between euthymic and manic states. No correlation between BDNF levels and YMRS nor HDRS scores. No correlation between BDNF levels and age nor in terms of length of illness. Higher levels of BDNF in patients suffering the disorder for more than 10 years. No differences in BDNF levels according to the presence of any psychiatric comorbidity.
3.	Barbosa et al. (2011)	49 BD-I patients (30 manic, 19 euthymic)	46.85 ± 12.56 (48.03 ± 13.66 manic, 45.00 ± 10.84 euthymic)	40.81%/59.18% (40.00%/60.00% manic, 42.1%/57.90% euthymic)	Severity of manic symptoms Severity of depressive symptoms Phase of the disorder, comorbidity and other clinical features	YMRS HAMD-17 MINI-plus	NGF	No differences in NGF levels between manic and euthymic patients. Levels of NGF negatively correlated with YMRS scores. No association between NGF levels and HAMD scores. No association between NGF levels and age nor number of hospitalisations in BD patients. In all BD patients, NGF levels were positively correlated with the length of illness but not for each sub-group (euthymia and mania). Levels of NGF did not differ in BD patients categorized according to the presence of psychiatric comorbidities, nor dependence of substances or nicotine.
4.	Barbosa et al. (2012)	25 BD-I euthymic patients	50.88 ± 9.11	32%/68%	Cognitive function Executive function Comorbidity and other clinical features	MMSE FAB MINI	BDNF TNF-α sTNFR sTNFR2	TNF-α levels were positively correlated with inhibitory control (FAB score). No other significant correlations between the biomarker levels and the cognitive parameters were evaluated (FAB and MMSE scores). No differences in BDNF, TNF-α, sTNFR1 or sTNFR2 levels according to the presence of psychiatric comorbidities, substance dependence and previous episodes of depression or psychosis.
5.	Barbosa et al. (2014)	46 BD-I patients (23 euthymic, 23 manic)	49.74 ± 11.41 (50.39 ± 8.07 euthymic, 49.09 ± 14.76 manic)	26.09%/73.91% (17.39%/82.61% euthymic, 34.79%/65.21% manic)	Severity of manic symptoms Severity of depressive symptoms Phase of the disorder and other clinical features	YMRS HDRS-17 MINI-Plus	IL-33 sSt2	No significant differences in IL-33 or sST2 levels between BD patients in mania and in euthymia phase. No correlation between IL-33 or sST2 levels and clinical parameters such as HDRS scores, YMRS scores and length of illness in BD patients.
6.	Benedetti et al. (2017)	40 BD-I inpatients current major depressive episode	46.3 ± 13.85	32.5%/67.5%	Child abuse and neglect	CTQ	BDNF	Early trauma was negatively associated with BDNF levels in BD patients.
7.	Bonnin at al. (2019)	102 BD patients (47 psychoeducation [25 BD-I], 39 FR [40 BD-I], 16 TAU [14 BD-I])	40.95 ± 8.20 (35.2 ± 9.2 psychoeducation, 48.5 ± 6.3 FR, 39.5 ± 9.9 TAU)	BD 63.20%36.79% (53.2%/46.8% psychoeducation, 73%/27% FR, 68.7%/31.3% TAU)	Severity of depressive symptoms Psychosocial functioning Clinical features	HAMD-17 FAST SCID	BDNF	Significant interaction between FAST scores at baseline and time on BDNF levels in BD patients. No significant interaction between age, HAMD scores and chronicity at baseline and time on BDNF levels.
8.	Brietzke et al. (2009)	61 BD-I patients (14 euthymic, 23 manic episode, 24 depressive episode)	42.23 ± 13.00 (44.2 ± 13.75 euthymic, 40.8 ± 13.70 manic, 45.0 ± 11.91 depressive)	39.33%/60.65% (28.6%/71.4% euthymic, 47.8%/52.2% manic, 37.5%/62.5% depressive)	Severity of manic symptoms Severity of depressive symptoms Phase of the disorder and other clinical features	YMRS HDRS SCID-I	IL-2 IL-4 IL-6 IL-10 IFN-γ TNF-α	No differences in levels of IL-2, IL-4, IL-6, IL-10, IFN-γ or TNF-α among euthymic, manic, depressive states. Positive correlation between IL-6 levels and YMRS and HDRS scores, and positive correlation between IL-2 levels and YMRS scores. No correlation between other biomarkers and the severity of depressive and manic symptoms. No correlation between any biomarker level and length of illness.
9.	Cetin et al. (2012)	45 BD-I patients in euthymic state (22 subsyndromal symptoms, 23 without subsyndromal symptoms)	35.71 ± 7.15 (36.86 ± 7.03 subsyndromal, 34.61 ± 7.28 without subsyndromal)	44.47%/55.52% (45.5%/54.5% subsyndromal, 43.5%/56.5% without subsyndromal)	Clinical characteristics and illness course Presence of subsyndromal symptoms	SKIP-TURK procedure	sIL-6R sTNF-R1	No differences in sTNF-R1 or sIL-6R levels between BD subsyndromal and without subsyndromal symptoms. No significant association between sTNF-R1 nor sIL-6R levels and length of illness in bipolar patients.
10.	Chang et al. (2017)	91 BD patients	43.0 ± 12.2	42.9%/57.1%	Dietary intake	BMI Daily caloric intake (kcal) Dietary LA (g)	IL-1β IL-1RA IL-6 IL-6RA IL-10 IL-18 IL-18BP sTNFR2	Dietary Linoleic Acid (LA) intake was inversely associated with IL-18BP levels. Positive association between sTNFR1 and sTNFR2 levels and BMI. No association between IL-1β, IL-6, IL-6RA, IL-10 or IL-18 levels and BMI, daily caloric intake and dietary Linoleic Acid. Positive association between IL-18, IL-18BP, STNFR1 and sTNFR2 levels and age.
11.	Chiou & Huang (2019)	83 BD-I patients (61 mania, 22 depression)	36.8 ± 11.4	45.78%/54.21%	Severity of manic symptoms Severity of depressive symptoms	YMRS HDRS-17	BDNF	Significant positive correlations between BDNF levels and YMRS scores among bipolar manic patients. No significant correlation between BDNF levels and HDRS scores among bipolar depressed patients.
12.	Dell’Osso et al. (2010)	33 BD outpatients current major depressive episode (17 unipolar, 16 BD-I)	46.4 ± 14.3 (22–65 years) (46.06 ± 10.78 unipolar, 46.75 ± 17.67 BD-I)	36.37%/63.63% (23.5%/76.5% unipolar, 50%/50% BD-I)	Severity of depressive symptoms Global illness severity	HRSD-21 CGI-S	BDNF	Significant negative correlation between BDNF levels and HDRS scores and between BDNF and CGI score in all patients and in the bipolar depressed group.
13.	Dolsen et al. (2018)	22 BD-I euthymic patients (11 CBT for insomnia, 11 psychoeducation)	36.4 ± 11.85	54,5%/45,5%	Severity of manic symptoms Severity of depressive symptoms Sleep disorders	YMRS, QIDS, DSISD, Sleep diary (TST and TWT)	IL-6 TNF-R2	Levels of IL-6 were positively correlated with severity of depressive symptoms (YMRS and QIDS scores), and inversely correlated with total sleep time but not with total wake time. No association between TNF-R2 and manic or depressive symptoms, neither with sleep disorders.
14.	Du et al. (2017)	48 BD patients (typical BD group) 63 BD with anxiety disorders patients (atypical BD group)	33.06 ± 14.53 BD 34.33 ± 15.55 atypical BD	31.3%/68.8% BD 28.6%/71.4% atypical BD	Comorbid diagnosis of anxiety disorders (atypical BD group)	ICD-10 criteria for anxiety disorders	IL-6 IL-8 IL-10 TNF-α	Levels of IL-6 and TNF-α were significantly higher in the atypical group (with anxiety) than in the BD group. The IL-10 levels were significantly lower in the atypical group than in the BD group. No differences in IL-8 levels between the BD group and the atypical group.
15.	Grande et al. (2014)	115 BD patients (79 BD-I, 18 BD-II, 18 NOS)	44.0 ± 20.0 (18–65 years)	29.6%/70.4%	Classification patients into early-stage and late-stage	SCID-I	BDNF IL-6	Differences in IL-6 levels between early-stage and late-stage. No differences in BDNF levels between early-stage and late-stage BD patients.
16.	Hope et al. (2015)	111 BD patients (65 BD-I, 40 BD-II, 6 NOS)	33 ± 10 (18–63 years)	54%/46%	General cognitive abilities	WASI, two subtests for verbal cognition and two tests for performance abilities	sTNF-R1 IL-1Ra sCD40L	Significant association between general cognitive abilities and levels of IL-1Ra and sCD40L, but not with sTNF-R1 levels.
17.	Isgren et al. (2015)	121 BD patients (65 BD-I, 46 BD-II, 10 NOS)	36.0 ± 22.0	38.8%/61.2%	Severity of manic symptoms Severity of depressive symptoms Global illness severity Alcohol and substance abuse Subtype, comorbidity and other clinical features	YMRS MADRS CGI AUDIT DUDIT MINI	IL-8	No association between changes of IL-8 levels and sex, bipolar subtype, CGI, MADRS and YMRS scores, length of illness, episode density, previous psychotic episodes, comorbid anxiety syndrome, attention deficit/hyperactivity disorder, smoking, alcohol abuse, substance abuse nor BMI. Levels of IL-8 were positively associated with age.
18.	Jacoby et al. (2016)	60 BD-I patients hospitalized with a manic or mixed episode	42.7 ± 11.7 (20–60 years)	61.7%/38.3%	Phase of the disorder	SCAN	BDNF TNF-α IL-6 IL-18	No differences in levels of BDNF, IL-6 or TNF-α between any affective states (euthymic, manic, depressive, or mixed). Differences in levels of IL-18 between mixed states and remission and between mixed and hypomanic/manic states (no significant in the adjusted model).
19.	Karabulut et al. (2019)	107 BD patients (77 BD chronic [50 euthymic state, 20 manic state, 7 depressive state], 30 BD early-stage patients [12 euthymic state, 13 manic state, 2 depressive state, 2 hypomanic state, 1 mixed state])	34.29 ± 10.24 (37.81 ± 11.9 BD chronic, 25.27 ± 6 BD early-stage)	52.34%/47.65% (57.2%/42.8% BD chronic, 23.4%/76.6% BD early-stage)	Severity of manic symptoms Severity of depressive symptoms Positive and negative symptoms Global illness severity Phase of the disorder, classification patients into early-stage and late-stage, comorbidity and other clinical features	YMRS MADRS PANSS CGI-S SCID-I	IL-1RA IL-6 TNF-α	Levels of IL-1RA, IL-6 and TNF-α were higher in chronic than early-stage patients. No differences in IL-1RA, IL-6 or TNF-α levels between chronic and early-stage patients in terms of mood state (euthymic, depressive and manic). Correlation between TNF-α levels and MADRS scores in chronic patients. Correlation between TNF-α levels and CGI scores in early-stage patients. No correlation between IL-1RA, IL-6 or TNF-α levels and YMRS nor PANNS scores among chronic and early-stage patients. No significant correlations between biomarkers levels and length of illness and the presence of psychotic symptoms.
20.	Kauer-Sant’Anna et al. (2007)	163 BD-I and BD-II patients (78 with life-time exposure to a traumatic event, 85 without-lifetime exposure to a traumatic event)	42.58 ± 11.58 (42.13 ± 12.00 presence of trauma, 43.01 ± 11.21 absence of trauma)	28.18%/71.82% (25.0%/75% presence of trauma, 31.1%/68.9% absence of trauma)	Trauma Comorbidity and other clinical features	DSM-IV A1 and A2 criteria from the PTSD module of the SCID SCID-I	BDNF	Patients with a history of trauma, especially sexual abuse, had lower levels of BDNF. No differences in BDNF levels in rapid cycling patients, or between patients with or without comorbid posttraumatic stress disorder (PTSD), alcohol abuse or dependence, or history of suicide attempts.
21.	Kenna et al. (2014)	47 BD euthymic patients (13 BD-I, 20 BD-II, 14 NOS)	32.97 ± 6.5 (18–45 years) (32.9 ± 6.4 BD-I, 32.1 ± 6.3 BD-II, 34.3 ± 6.9 NOS)	0%/100%	Severity of depressive symptoms Subtype	MADRS SCID	BDNF	No significant differences in BDNF levels between women with BD-I, BD-II and NOS. A significant negative correlation between BDNF levels and MADRS scores in BD women patients.
22.	Kim et al. (2013)	116 BD-I patients in manic episode, with pharmacological treatment during 6 weeks of follow-up	35.90 ± 11.76 (35.99 ± 11.67)	36.2%/63.8% (37.3%/62.7%)	Severity of manic symptoms	YMRS	IGF-1 β-NGF BDNF	No correlation between IGF-1, β-NGF or BDNF levels and YMRS scores (after 6 weeks of pharmacological treatment).
23.	Lee et al. (2014)	232 BD-II patients in depressed state, with VPA treatment during 12 weeks of follow-up (115 VPA + Memantine, 117 VPA + placebo)	31.77 ± 11.56	50.86%/49.13%	Severity of manic symptoms Severity of depressive symptoms	YMRS HDRS	TNF-α IL-1β IL-6 IL-8	The changes in IL-6 and IL-1β levels were significantly associated with the changes in HDRS scores (before and after pharmacological treatment). The changes in TNF-α levels were significantly associated with the changes in YMRS scores (before and after pharmacological treatment). The changes in IL-8 levels were not associated with the changes in HDRS scores, nor with the changes in YMRS scores (before and after pharmacological treatment).
24.	Lee et al. (2016)	541 BD patients (117 BD-I, 424 BD-II), with add-on memantine/placebo/DM treatment during 12 weeks of follow-up	31.2 ± 11.2	48.79%/51.20%	Cognitive function Quality of life	WCST and CPT WHOQOL-BREF, Chinese version	BDNF	Only in BD-I patients was there a significant negative correlation between changes in BDNF levels and cognitive function (WCST scores) (after 12-weeks of pharmacological treatment) (not after a correction for multiple comparison). No correlation between changes of BDNF levels and CPT scores nor quality of life (WHOQOL scores).
25.	Loch et al. (2015)	23 BD patients in depressive episode (8 BD-I, 15 BD-II), with litium treatment during 6 weeks of follow-up	28.0 (18–43 years)	17.4%/82.6%	Severity of depressive symptoms Subtype, comorbidity and other clinical features	HAMD-21 SCID	NT-3 NT-4/5	No correlation between NT-3 or NT-4/5 levels and changes in HAMD scores (after 6 weeks of treatment). No association between clinical features such as bipolar subtype, psychotic symptoms or tobacco use and baseline NT levels, except length of illness, which showed a positive correlation with NT-3 and NT-4/5 levels.
26.	Machado-Vieira et al. (2007)	30 BD-I patients in mania state (22 drug-naïve, 8 drug-free)	26 ± 4 (20–40 years)	23.3%/76.7%	Severity of manic symptoms	YMRS	BDNF	Significant negative correlation between BDNF levels and YMRS scores in unmedicated BD patients.
27.	Maiti et al. (2017)	25 BD patients current episode mania, 21 with oxcarbazepine treatment during 4 weeks of follow-up	34.16 ± 9.89 (18–45 years)	80%/20%	Severity of manic symptoms	YMRS	BDNF	Inverse relationship between BDNF levels and the YMRS score at baseline, and positive correlation between change of BDNF and change in the YMRS score (after four weeks of treatment).
28.	Mansur et al. (2020)	55 BD I-II patients current major depressive episode (27 Infliximab + 28 placebo)	44.91 ± 10.90 (44.04 ± 11.55 Infliximab, 45.75 ± 10.28 placebo)	20%/80% (25.9%/74.1% Infliximab, 14.3%/85.7% placebo)	Severity of depressive symptoms Child abuse and neglect Clinical features	MADRS CTQ and the subdomain of PA MINI 5.0.0 for DSM-IV-TR	TNFR1 TNFR2	Levels of TNFR1 were significantly associated with MADRS scores, but there was no association between TNFR2 levels and MADRS scores. Levels of TNFR1 and TNFR2 were not associated with CTQ scores. No association between TNFR1 and TNFR2 and age, sex, BMI or tobacco use at baseline.
29.	Mizuno et al. (2016)	60 BD patients	50.2 ± 13.8 (19–75 years)	46.7%/53.3%	Resilience	Resilience Scale, 25-item	BDNF	No significant correlation between BDNF levels and resilience in the BD group.
30a.	Mondin et al. (2016)	48 BD patients (10 mania/mixed episode, 27 depressive episode, 11 euthymic)	21.92 ± 2.32 (18–24 years)	25%/75%	Chronotype	BRIAN	IL-6 IL-10 TNF-α	Association between subjects with day/night cycle reverse and IL-6 levels. BD patients active at night had IL-6 levels that were significantly decreased in comparison with BD patients who were not active at night. The reversed day/night cycle BD patients had IL-6 levels that were decreased in comparison with non-reversed day/night cycle BD patients.
30b.	Wiener et al. (2017)	48 BD patients	21.92 ± 2.32 (18–24 years)	25%/75%	Severity of depressive symptoms Functional impairment	HDRS FAST	IL-6 IL-10	No correlation between IL-6 or IL-10 levels and HDRS scores in BD patients. Positive correlation between IL-6 and IL-10 levels and functional impairment (FAST scores). Higher IL-6 levels were associated with higher levels of impairment in the cognitive domain and in interpersonal relationships. Levels of IL-10 were positively correlated with impairment in the cognitive domain and in occupational functioning.
31.	Monteleone et al. (2008)	28 BD euthymic patients (17 BD-I, 11 BD-II)	45.10 ± 11.08 (46.6 ± 9.4 BD-I, 42.8 ± 13.7 BD-II)	39.28%/60.70% (29.41%/70.58% BD-I, 54.54%/45.45% BD-II)	Subtype and other clinical features	SCID-IP, Patient Edition	BDNF	No significant differences in BDNF levels among patient groups (BD-I, BD-II). No significant correlations between BDNF levels and age, age of onset, length of illness and number of depressive episodes in each diagnostic BD group.
32.	Mora et al. (2019)	84 BD patients (52 euthymic, 32 manic)	45.13 ± 12.28 (18–65 years) (47.52 ± 11.9 euthymic BD, 41.25 ± 12.9 manic)	52.38%/47.62% (50%/50% euthymic, 56.3%/43.7% manic)	Intelligence Attention Memory Executive function Subtype, phase of the disorder and other clinical features	WAIS III (vocabulary, block design and digits subtests), WCST, Stroop Color and Word Test, FAS verbal fluency task, TMT, CPT-II, CVLT, RCFT SCID-I	BDNF IL-6 IL-10 TNF-α	No differences in BDNF, IL-6, IL-10 or TNF-α levels between BD-I and BD-II euthymic patients. No differences between euthymic and manic patients in BDNF, IL-6, IL-10 or TNF-α levels. In a regression model, BDNF was the only biomarker associated with executive functioning and predicted worse performance in verbal memory in BD patient groups. Levels of IL-6 were not associated with cognitive domains in BD patients. Only IL-6 levels were associated with length of illness in BD manic and euthymic patients.
33a.	Munkholm et al. (2014)	37 BD rapid cycling patients (22 BD-I, 15 BD-II)	40.9 ± 12.3 (18–70 years)	32.43%/67.56%	Severity of manic and depressive symptoms Phase of the disorder and other clinical feature	YMRS HAMD-17 SCAN	BDNF	No difference in BDNF levels regardless of mood state (euthymic, depressive, manic/hypomanic, mixed). No association between BDNF levels and HAMD scores in a depressive state, neither between BDNF levels and YMRS scores in a manic/hypomanic state. Higher BDNF levels in patients with more than 10 years of illness in comparison with a shorter duration of disease.
33b.	Munkholm et al. (2015)	37 BD rapid cycling patients (22 BD-I, 15 BD-II)	40.9 ± 12.3 (18–70 years)	32.43%/67.56%	Severity of manic symptoms Severity of depressive symptoms Phase of the disorder and other clinical features	YMRS HAMD-17 SCAN	IL-6 IL-18	Higher levels of IL-6 and IL-18 in manic/hypomanic rather than in depressive and euthymic patients. No differences in IL-6 or IL-18 levels between a depressed and euthymic state. No association between IL-6 and IL-18 levels and HAMD scores in depressive patients. No association between IL-6 and IL-18 levels and YMRS scores in manic/hypomanic patients. In mixed states, IL-6 levels were higher in episodes of 1–2 weeks compared with episodes below one week. In manic/hypomanic states, IL-6 levels were lower in episodes of more than one month compared with episodes below one week. In manic/hypomanic states, IL-18 levels were higher in episodes of 1–2 weeks compared with episodes below one week. In depressive states, IL-18 levels were lower in some episodes compared with episodes below one week. In euthymic states, IL-18 levels were higher in episodes of 3–4 weeks compared with episodes below one week.
34.	Ortiz-Dominguez et al. (2007)	20 BD-I patients (10 manic episode, 10 depressive episode)	34.3 ± 9.94 (20–50 years) (28.9 ± 8.45 manic, 39.7 ± 11.43 depressive)	25%/75% (30%/70% manic, 20%/80% depressive)	Phase of the disorder	MINI for DSM-IV	TNF-α IL-1β IL-2 IL-4 IL-6	Bipolar patients in a manic state showed a significant increase in IL-4 levels and a significant decrease in IL-1β and IL-6 levels compared with depressed state patients. No significant differences between bipolar patients in a depressive state and in a manic state in IL-2 or TNF-α levels.
35.	Pantović-Stefanović et al. (2016)	83 BD-I patients, with pharmacological treatment during 10 weeks of follow-up	45.61 ± 11.05	36.4%/63.60%	Severity of manic symptoms Severity of depressive symptoms Global illness severity Cardinal features of the disorder Clinical features	YMRS MADRS CGI-BP-S BPIX SCID-I	sICAM-1 sVCAM-1 IL-6 TNF-α	No differences in sICAM-1, sVCAM-1, IL-6, or TNF-α between patients in an acute phase and in remission phase. Levels of sVCAM-1 correlated negatively with the Bipolarity Index in the acute and remission phase. BD patients with manic polarity had significantly lower levels of sVCAM-1 compared with BD patients with depressive polarity. There were no differences regarding the polarity of the acute episode and levels of sICAM-1, IL-6 or TNF-α. Levels of sVCAM-1 in acute phase had a significant negative correlation with YMRS and a positive correlation with MADRS scores. Levels of sVCAM-1 correlated positively with the severity of the acute episode based on CGI-BP-S score. No correlation between sICAM-1, IL-6 or TNF-α levels and CGI, MADRS and YMRS scores in an acute or remission phase. Levels of sVCAM-1 in the acute and remission phase were positively correlated with the age of onset. Levels of sVCAM-1 in the acute phase were negatively correlated with the number of previous psychotic episodes. There were significantly lower levels of sVCAM-1 in the acute phase in patients with a family history of BD. Levels of TNF-α were inversely correlated with the duration of depressive episodes in BD patients in the acute phase. Significantly lower TNF- α levels in acute and remission phase in patients with a longer duration of untreated BD. No correlation between sICAM-1 nor IL-6 levels and clinical features.
36.	Quidé et al. (2018)	69 BD-I patients	38.11 ± 12.31 (18–65 years)	33.33%/66.66%	Childhood trauma	CTQ	IL-6 TNF-α	No association between IL-6 nor TNF-α levels and CTQ domains (emotional abuse, physical abuse, sexual abuse, emotional neglect and physical neglect) in the BD group.
37.	Reininghaus et al. (2015)	101 BD euthymic patients (64 BD-I, 37 BD-II)	44.00 ± 12.99	48.51%/51.48%	Weigh cycling Clinical features	WCYC SCID-I	IL-6	Significant differences in levels of IL-6 between weight cycling (WCY), BD patients and non-weight cycling (non-WCY) BD patients, WCY showed higher IL-6 levels than non-WCY. Significative correlation between IL-6 levels and the number of depressive episodes in BD patients.
38.	Rosa et al. (2006)	44 BD-I patients (15 manic, 14 depressed, 15 euthymic)	40.83 ± 9.29 (30–51 years) (40.1 ± 9.3 manic, 42.1 ± 8.2 depressed, 40.4 ± 10.3 euthymic)	41.07%/58.95% (56.3%/43.8% manic, 28.6%/71.4% depressed, 37.5%/62.5% euthymic)	Severity of manic symptoms Severity of depressive symptoms	YMRS HRSD-17	GDNF	Higher GDNF levels in manic and depressed BD patients as compared with euthymic patients.
39a.	Siwek et al. (2016)	133 BD patients (65 BD-I and 64 BD-II) (23 manic phase, 61 depressive phase, 49 euthymic) (35 with melancholia, 26 without melancholia)	44.3 ± 12.9 (21–70 years)	65.41%/34.58%	Severity of manic symptoms Severity of depressive symptoms Subtype, phase of the disorder, comorbidity and other clinical features	YMRS MADRS HDRS SCID-I	sIL-1RA sIL-2R sIL-6R, sTNFR60 sTNFR80	No differences in sIL-1RA levels regarding the mood phase (manic, depressive, or euthymic). Levels of sTNFR80 were significantly higher in depressive episodes in comparison with a (hypo)manic episode and remission. Levels of sTNFR80 were significantly higher in melancholia in comparison with all other patients (without melancholia). Positive correlation between sTNFR60 and sTNFR80 levels and HDRS and MADRS scores. No significant association between any biomarker and YMRS scores. No significant association between any biomarker and clinical variables such as subtype, number of lifetime episodes, age of onset, length of illness, rapid cycling, atypical depression, suicidal ideation, and psychotic symptoms.
39b.	Sowa-Kućma et al. (2017)	133 BD patients (69 BD-I, 64 BD-II all in depressed or euthymic phase)	44.3 ± 12.9 BD (42.0 ± 14.6 BD-I, 46.8 ± 10.2 BD-II)	65.41%/34.58% (42.02%/57.97% BD-I, 26.56%/73.43% BD-II)	Subtype and other clinical features	SCID-I	sIL-1RA IL-1α sIL-2R sIL-6R sTNFR60 sTNFR80	No significant differences in any of the biomarkers between BD-I and BD-II. Levels of IL-1α and sTNFR80 were increased in BD with melancholia in comparison with BD without melancholia.
40.	Soares et al. (2016)	118 BD patients in euthymic state (91 BD-I, 27 BD-II)	64.0 ± 9.7 (64.2 ± 9.8 BD-I, 63.3 ± 9.5 BD-II)	31.29%/68.70% (30.7%/69.3% BD-I, 33.3%/66.7% BD-II)	Severity of manic and depressive symptoms Subtype and other clinical features	YMRS HAMD-17 SCID-I	BDNF	Patients with BD-I had significantly lower BDNF levels than patients with BD-II. No significant correlation between BDNF levels and age, length of illness, YMRS nor HAMD scores.
41.	Tatay-Manteiga et al. (2017)	48 BD euthymic patients (25 early-stage, 23 late-stage)	44.21 ± 10.06 (18–60 years) (43.4 ± 10.3 early-stage, 45.1 ± 9.8 late-stage)	47.91%/52.08% (48%/52% early-stage, 47.82%/52.17% late-stage)	Classification patients into early-stage and late-stage	Clinical interview	BDNF TNF-α IL-6 IL-10 NT-3	Levels of IL-10 were significantly higher in early-stage bipolar patients compared with late-stage. There were no significant differences in BDNF, TNF-α, IL-6 or NT-3 levels between early-stage and late-stage bipolar patients.
42.	Tunca et al. (2014)	96 BD patients (92 BD-I, 4 BD-II) (37 euthymic, 33 manic, 26 depressed)	38.12 ± 10.84 (36.24 ± 10.02 euthymia, 36.77 ± 12.02 mania, 42.54 ± 10.52 depression)	44.74%/55.25% (37.8%/62.2% euthymia, 34.3%/65.7% mania, 67.9%/32.1% depression)	Severity of manic and depressive symptoms Global illness severity Social, occupational, and psychological functioning Phase of the disorder and other clinical features	YMRS HDRS CGI GAF SCID-I	BDNF GDNF	There were significantly lower BDNF levels in depressive and manic bipolar patients than in euthymic patients, but there were no differences in BDNF levels between manic and depressive episodes. Neither BDNF nor GDNF levels correlated with the duration of the mood state (euthymia, manic, depressive). There was a negative correlation between BDNF levels and YMRS, HDRS and CGI scores. There was a positive correlation between BDNF levels and GAF scores. Therew as a negative correlation between GDNF levels and HDRS scores and no correlation between GDNF levels and YMRS, CGI nor GAF scores. There were higher GDNF levels in early onset rather than late onset BD patients, while BDNF levels were similar in both groups.
43.	Uyanik et al. (2015)	30 BD-I patients in manic episode, with pharmacological treatment during 6 weeks of follow-up	33.4 ± 8.6 (18–65 years)	53.3%/46.7%	Severity of manic symptoms	YMRS	IL-4 IL-6 IL-10 TNF-α IFN-γ	No correlation between IL-4, IL-6, IL-10, TNF-α nor IFN-γ levels and YMRS scores (before and after pharmacological treatment).
44a.	van den Ameele et al. (2017)	67 BD patients (35 depressive episode [16 BD-I, 19 BD-II], 32 (hypo)manic episode [26 BD-I, 4 BD-II, 2 schizoaffective])	43.3 ± 11.1 (23–62 years) (43.7 ± 9.7 (28–61 years) depressive, 42.9 ± 12.7 (23–62 years) (hypo)manic)	41.8%/58.2% (31.4%/68.6% depressive 53.1%/46.9% (hypo)manic)	Severity of manic symptoms Severity of depressive symptoms Positive symptoms Subtype, phase of the disorder, comorbidity and other clinical features	YMRS HDRS-17 PANSS, positive subscale MINI-plus, version 5.0.0	TNF-α BDNF VEGF sFlt-1	In a subgroup analysis, patients with mood symptoms had a significant increase in TNF-α levels in comparison with euthymic patients, but there were no significant differences between euthymic patients and patients with mood symptoms in BDNF, VEGF and sFlt-1 levels. No strong correlation between TNF-α, BDNF, VEGF and sFlt-1 levels and HDRS, YMRS nor PANSS positive subscale scores in BD patients. There was no significant association between TNF-α, BDNF, VEGF and sFlt-1 levels and clinical characteristics such as BD type I or II, mixed features, current psychotic features, lifetime psychosis features, age of onset, length of illness nor number of hospitalizations. There were higher concentrations of all the biomarkers levels, and this effect was significant for TNF-α, VEGF and sFlt-1, when the duration of the current mood episode was shorter.
44b.	van den Ameele et al. (2018)	67 BD patients (42 BD-I, 23 BD-II, 2 schizoaffective disorder) (29 depressive episode, 29 (hypo)manic episode, 9 mixed episode)	43.3 ± 11.1 (23–62 years)	41.8%/58.2%	Severity of manic symptoms Severity of depressive symptoms Positive symptoms Subtype, phase of the disorder, comorbidity and other clinical features	YMRS HDRS-17 PANSS, positive subscale MINI-plus	TNF-α IFN-y IL-6	No significant differences were found between levels of TNF-α, IFN-y or IL-6 and YMRS, HDRS, nor between PANSS scores in BD patients. No significant relations were found between any of the biomarkers or illness characteristics such as BD type I or II, mood state, duration of current mood state, number of mood episodes and number of hospitalizations, nor lifetime psychotic features. A longer duration of illness was associated with higher TNF-α. The older patient group (above 45 years old) had higher levels of IL-6 and TNF-α.
45.	Wang et al. (2016)	48 BD-II patients (placebo) 41 SBD (more than 2-days but less than 4-days duration of hypomania) patients (placebo)	31.5 ± 11.3 BD-II 28.5 ± 10.2 SBD	54.16%/45.83% BD-II 51.21%/48.78% SBD	History of subthreshold hypomania (SBD group)	SADS-L, Chinese version	TNF-α TGF-β 1 IL-6 IL-8 IL-1β BDNF	No significant differences were found in TNF-α, TGF-ß1 or IL-8 levels between the BD group and the SBD group at baseline. Significantly higher levels of IL-1ß but significantly lower levels of IL-6 and BDNF in the SBD group were found at baseline. Levels of BDNF were significantly lower in the SBD group (after 12 weeks of treatment). No other significant differences in biomarker levels between the BD and SBD group (after 12 weeks of treatment) were found.
46.	Wang et al. (2016)	737 BD patients (234 BD-I, 260 BD-II, 243 SBD)	32.69 ± 12.20 (33.6 ± 11.7 BD-I, 31.6 ± 12.3 BD-II, 33.0 ± 12.6 SBD)	47.48%/52.50% (49.14%/50.85% BD-I, 51.92%/48.07% BD-II, 41.15%/58.84% SBD)	Severity of manic and depressive symptoms Subtype, subthreshold hypomania (SBD group), comorbidity and other clinical features	YMRS HDRS SADS-L, Chinese version	BDNF TNF-α TGF-ß1 IL-8	BDNF levels were not associated with bipolar patient groups (BD-I, BD-II, SBD). BD-II and SBD patients had lower levels of IL-8 than BD-I patients.There were no significant differences in TNF-α, TGF-ß1, or IL-8 levels between BD-II and SBD patients. There was a significant association between TNF-α levels and HDRS scores and between TGF-ß1 levels and HDRS and YMRS scores, length of illness and comorbidities.
47.	Yoshimura et al. (2006)	18 BD-I patients (12 manic episode, 6 depressive episode), with risperidone treatment during 4 weeks of follow-up	34 ± 15 (23–51 years)	44.44%/55.55%	Phase of the disorder	Clinical interview	BDNF	Levels of BDNF were significantly decreased in bipolar depressive patients compared with manic patients (before and after pharmacological treatment).

Abbreviations: AUDIT, Alcohol Use Disorders Identification Test; BD, Bipolar Disorder; BMI, Body Mass Index; BPIX, Bipolarity Index; BRIAN, Biological Rhythm Interview of Assessment in Neuropsychiatry; CBT, Cognitive Behavioral Therapy; CGI-BP-S, Clinical Global Impression Scale of Bipolar Disorder-Severity; CGI-S, Clinical Global Impression Severity of Illness Scale; CPT, Conners’ Continuous Performance Test; CPT-II, Conners’ continuous Performance Test II; CVLT, California Verbal Learning Test; DSISD, Duke Structured Interview for Sleep Disorder; DSM, Diagnostic and Statistical Manual of Mental Disorders; DUDIT, Drug Use Disorders Identification Test; FAB, Frontal Assessment Battery; FAST, Functioning Assessment Short Test; FR, Functional Remediation; GAF, Global Assessment of Functioning; HC, Healthy Controls; HDRS, Hamilton Depression Rating Scale; ICD, International Classification of Diseases; LA, Linoleic Acid; MADRS, Montgomery-Åsberg Depression Rating Scale; MINI, Mini-International Neuropsychiatric Interview; MMSE, Mini-Mental State Examination; NOS, Not Otherwise Specified; PA, Physical Abuse; PANNS, Positive and Negative Syndrome Scale; PTSD, Posttraumatic Stress Disorder; QIDS, Quick Inventory of Depressive Symptomatology; RCFT, Rey-Osterrieth Complex figure; SADS-L, Structured Interview of the Modified Schedule of Affective Disorder and Schizophrenia-Life Time; SBD, Subthreshold Bipolar Disorder; SCAN, The Schedules for Clinical Assessment in Neuropsychiatry; SCID-I, Structured Clinical Interview for DSM-IV, Axis I Disorders; TAU, Treatment as usual; TMT, Trail Making Test; TST, Total Sleep Time; TWT, Total Wake Time; VPA, Valproate; WAIS III, Wechsler Adult Intelligence Scale; WASI, Wechsler Abbreviated Scale of Intelligence Scale; WCST, Wisconsin Card Sorting Test; WCYC, Weight Cycling Classification of the Nurses Health Study; WHOQOL-BREF, Brief version of the World Health Organization Quality of Life instrument; YMRS, Young Mania Rating Scale.

## Data Availability

Not applicable.

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
