# Peer review of "Clinical Value of Inflammatory and Neurotrophic Biomarkers in Bipolar Disorder: A Systematic Review and Meta-Analysis"

_biomedicines, 2022, doi:10.3390/biomedicines10061368_

Round 1
Reviewer 1 Report
The authors reviewed findings related to potential application of inflammatory and neurotrophic biomarkers in distinguishing various clinical parameters of bipolar disorder (subtypes, phases, severity, episodes,…). They used Preferred Reporting Items for Systematic Review and Meta-Analyses (PRISMA) (PROSPERO #CRD42020180626). The main finding is a negative correlation between circulating BDNF levels and depression severity score.
The study was performed correctly and it gives the comprehensive review of the current knowledge in the field. The manuscript is clearly written and presented, although the entire text is not interesting to read, it is hard to see the practical clinical application of the work presented and ultimately the manuscript does not bring an attractive/interesting conclusion. It is an old topic that BDNF decreases in depressed patients.
As the manuscript is focused on inflammatory mediators and BDNF, I would suggest to briefly comment the contribution of cytokine- and BDNF-related genetic and epigenetic factors (if any) in the Introduction section.
In the last paragraph of the Introduction, the authors wrote: “The effort to structure and expand knowledge in this area of study has important implications for improving the objectivity of the diagnosis and the validity of treatments.“ In that light, please comment the practical clinical value of the work presented. How exactly the main finding (and biomarkers in general) can improve the objectivity of diagnosis and the validity of treatments? From the Introduction, it is even not evident that some challenges exist related to diagnosis and treatment.
As it seems, based on the current data, the inflammatory mediators and BDNF are not useful biomarkers in BD. Please put forward other potential targets whose contribution to BD should be investigated in the future (in the Discussion).
Author Response
The authors reviewed findings related to potential application of inflammatory and neurotrophic biomarkers in distinguishing various clinical parameters of bipolar disorder (subtypes, phases, severity, episodes,…). They used Preferred Reporting Items for Systematic Review and Meta-Analyses (PRISMA) (PROSPERO #CRD42020180626). The main finding is a negative correlation between circulating BDNF levels and depression severity score.
- The study was performed correctly and it gives the comprehensive review of the current knowledge in the field. The manuscript is clearly written and presented, although the entire text is not interesting to read, it is hard to see the practical clinical application of the work presented and ultimately the manuscript does not bring an attractive/interesting conclusion. It is an old topic that BDNF decreases in depressed patients.
Response: We thank the reviewer for the comments. We have revised the entire manuscript to make it more interesting reading in terms of the clinical application of the study. The conclusion has also been revised. One of the main conclusion of the present review is that the known evidence for BDNF down-regulation and inflammatory up-regulation in major depression remains unclear or inconsistent in patients with BD. Some of these inconsistencies are based on a number of important uncontrolled confounding factors, such as end-point measurements, a priori non-relevant parameters (BMI, daily caloric intake), certain clinical interviews that hamper the reliability of the results, and different populations with respect to age, sex, metabolic alterations, cognitive function and symptom severity that are not accounted for by BD studies.
- As the manuscript is focused on inflammatory mediators and BDNF, I would suggest to briefly comment the contribution of cytokine- and BDNF-related genetic and epigenetic factors (if any) in the Introduction section.
Response: Introduction section has been improved following the reviewer recommendation. Regarding the identification of genetic markers for BD, candidate gene strategy has been largely unsuccessful. BDNF is one of the most investigated candidate genes by studies looking genetic (GWAS) and epigenetic (DNA methylation and EWAS) variations associated with the risk of BD. Evidence from longitudinal analysis of methylation profiles supports a role of inflammatory factors (e.g. BDNF, IL-1β) in the pathogenesis of BD. However, potential confounders (underpowered sample sizes, inadequate testing correction, undetected genetic mismatch, global methylation or inconclusive results compared to other psychiatric disorders) may explain the lack of consistent findings and the difficulties in establishing BD susceptibility genes. Please, see line 94. Additional references:
Gordovez FJA, McMahon FJ. The genetics of bipolar disorder. Mol Psychiatry. 2020 Mar;25(3):544-559. doi: 10.1038/s41380-019-0634-7
Duffy A, Goodday SM, Keown-Stoneman C, Scotti M, Maitra M, Nagy C, Horrocks J, Tu-recki G. Epigenetic markers in inflammation-related genes associated with mood disorder: a cross-sectional and longitudinal study in high-risk offspring of bipolar parents. Int J Bi-polar Disord. 2019 Aug 6;7(1):17. doi: 10.1186/s40345-019-0152-1
Alameda L, Trotta G, Quigley H, Rodriguez V, Gadelrab R, Dwir D, Dempster E, Wong CCY, Forti MD. Can epigenetics shine a light on the biological pathways underlying major mental disorders? Psychol Med. 2022 Feb 23:1-21. doi: 10.1017/S0033291721005559
- In the last paragraph of the Introduction, the authors wrote: “The effort to structure and expand knowledge in this area of study has important implications for improving the objectivity of the diagnosis and the validity of treatments.“ In that light, please comment the practical clinical value of the work presented. How exactly the main finding (and biomarkers in general) can improve the objectivity of diagnosis and the validity of treatments? From the Introduction, it is even not evident that some challenges exist related to diagnosis and treatment.
Response: The reviewer is correct. As stated in the Introduction section, objectivity of the diagnosis and the validity of treatment in patients with BD have become a main challenge of this clinical area of knowledge. The clinical value of inflammatory biomarkers has yet to be demonstrated by highly controlled studies that follow a precise phenotyping (type, phase, severity, and comorbidities) of BD patients in the time of end-point measurements. Despite the confounding factors stated above, the main finding of the present systematic review and meta-analysis is the negative correlation found between BDNF levels and severity of depressive symptoms in BD patients. This evidence suggests that BDNF may be an eventual biomarker for BD diagnosis and could be considered for the search of more efficient treatments in clinical trials. Please, see line 120.
- As it seems, based on the current data, the inflammatory mediators and BDNF are not useful biomarkers in BD. Please put forward other potential targets whose contribution to BD should be investigated in the future (in the Discussion).
Response: Discussion has been updated following reviewer’s comment. Prominent evidence from our review study suggests that BDNF may be an eventual biomarker for BD. This inconsistent finding is due to a number of confounders in uncontrolled studies of inflammatory factors that highlight the need for accurate biomarkers for BD progression. Several additional inflammatory factors, such as specific chemokines (MCP-1, fractalkine), glial factors (GFAP, S100B), or lipid-derived mediators such as endocannabinoids, and neurodegeneration-related molecules such as neurofilament light chain, neurogranin and β-amyloid peptide isoforms (Aβ42, Aβ40, Aβ38) and their fragments could be useful biomarkers in patients with BD in relation to prospective clinical outcomes, such as those of neuronal injury, cognitive deficits and risk of dementia. Recent studies have evaluated non-coding RNA expression, sexual hormones, oxidative stress (uric acid, bilirubin, albumin) and metabolic biomarkers (metabolic syndrome, overweight/obesity, thyroid and liver function) in predicting BD and suicidality in individuals with bipolar disorder. In addition, big data of cerebral blood flow and structural and functional connectivity by magnetic resonance imaging are becoming useful tools for real-world practice in psychiatry. Please, see line 479. Additional references:
Knorr U, Simonsen AH, Jensen CS, Zetterberg H, Blennow K, Akhøj M, Forman J, Hasselbalch SG, Kessing LV. Alzheimer's disease related biomarkers in bipolar disorder - A longitudinal one-year case-control study. J Affect Disord. 2022 Jan 15;297:623-633. doi: 10.1016/j.jad.2021.10.074.
Maloum Z, Taheri M, Ghafouri-Fard S, Shirvani-Farsani Z. Significant reduction of long non-coding RNAs expression in bipolar disorder. BMC Psychiatry. 2022 Apr 12;22(1):256. doi: 10.1186/s12888-022-03899-y.
Steinacker P, Al Shweiki MR, Oeckl P, Graf H, Ludolph AC, Schönfeldt-Lecuona C, Otto M. Glial fibrillary acidic protein as blood biomarker for differential diagnosis and severity of major depressive disorder. J Psychiatr Res. 2021 Dec;144:54-58. doi: 10.1016/j.jpsychires.2021.09.012.
Isgren A, Sellgren C, Ekman CJ, Holmén-Larsson J, Blennow K, Zetterberg H, Jakobsson J, Landén M. Markers of neuroinflammation and neuronal injury in bipolar disorder: Relation to prospective clinical outcomes. Brain Behav Immun. 2017 Oct;65:195-201. doi: 10.1016/j.bbi.2017.05.002.
Niu Z, Wu X, Zhu Y, Yang L, Shi Y, Wang Y, Qiu H, Gu W, Wu Y, Long X, Lu Z, Hu S, Yao Z, Yang H, Liu T, Xia Y, Chen Z, Chen J, Fang Y. Early Diagnosis of Bipolar Disorder Coming Soon: Application of an Oxidative Stress Injury Biomarker (BIOS) Model. Neurosci Bull. 2022 May 19. doi: 10.1007/s12264-022-00871-4.
Stenzel C, Dalkner N, Unterrainer HF, Birner A, Bengesser SA, Fellendorf FT, Fink A, Fleischmann E, Lenger M, Maget A, Platzer M, Queissner R, Schönthaler E, Tmava-Berisha A, Reininghaus EZ. Effects of metabolic syndrome and obesity on suicidality in individuals with bipolar disorder. J Affect Disord. 2022 May 14:S0165-0327(22)00576-6. doi: 10.1016/j.jad.2022.05.062.
Wei Y, Womer FY, Sun K, Zhu Y, Sun D, Duan J, Zhang R, Wei S, Jiang X, Zhang Y, Tang Y, Zhang X, Wang F. Applying dimensional psychopathology: transdiagnostic prediction of executive cognition using brain connectivity and inflammatory biomarkers. Psychol Med. 2022 May 10:1-11. doi: 10.1017/S0033291722000174.
Cattarinussi G, Kubera KM, Hirjak D, Wolf RC, Sambataro F. Neural Correlates of the Risk for Schizophrenia and Bipolar Disorder: A Meta-analysis of Structural and Functional Neuroimaging Studies. Biol Psychiatry. 2022 Mar 4:S0006-3223(22)01068-X. doi: 10.1016/j.biopsych.2022.02.960.
Reviewer 2 Report
This is a systematic review and meta-analysis of circulating inflammatory and neurotrophic factors as reliable biomarkers for clinical parameters (subtypes, phases, severity, episodes, among others) of BD. Although methodologically solid I think there some aspects that need to be tackled by the authors:
- The search might have missed relevant studies (PMID: 32919410)
- It is unclear why a meta-analysis was performed for variables with 3 or more studies, 2 could have been sufficient to perform a quantitative synthesis
- The rates of the most common markers should be reported
- Psychometric measures were different among studies, but it is not clear how these were standardized (I.e. severity of mood symptoms)
- The discussion is somewhat confusing, the authors state that “This result is consistent with one of the most replicated findings in the literature, the decreased peripheral levels of BDNF in BD”. However, this meta-analysis did not show a difference in depression and mania compared to euthymia.
Author Response
This is a systematic review and meta-analysis of circulating inflammatory and neurotrophic factors as reliable biomarkers for clinical parameters (subtypes, phases, severity, episodes, among others) of BD. Although methodologically solid I think there some aspects that need to be tackled by the authors:
- The search might have missed relevant studies (PMID: 32919410)
Response: It is possible we have overlooked some relevant studies, although we think we have conducted a very thorough search. The reason why we have not included the study (PMID: 32919410) is because the only variable analyzed in BD is the response to lithium (Exclusion criteria (g): studies in which the only exposure factor was a pharmacological treatment or psychological intervention). Moreover, we conducted the systematic review until May 2020 and the study was published for early view on September 2020.
- It is unclear why a meta-analysis was performed for variables with 3 or more studies, 2 could have been sufficient to perform a quantitative synthesis
Response: The reviewer is correct. Although meta-analysis is technically possible with 2 studies, we considered that these studies were at risk of bias because the studies were observational. In addition, the results were heterogeneous. We decided to perform meta-analysis with 3 or more studies in order to increase our statistical power by reducing the standard error of the weighted average effect size. We have included a sentence in the limitation section
- The rates of the most common markers should be reported
Response: We have calculated the rates of the 33 biomarkers analyzed across the 51 studies selected for the present systematic review. This information has been added to supplementary material (Table S2). The rates were [N(%)]: BDNF = 25(49%); IL-6 = 20(39.2%); TNF-α = 17(33.3%); IL-10 = 8(19.6%); IL-8 = 5(9.8%); IL-1β, sIL-6R and sTNFR1 = 4(7,8%); sIL-2R, IFNγ, NGF, IL-1Ra, IL-4 and IL-18 = 3(5.9%); TGF-β1, sTNFR2, sIL-1Ra, IL-2, GDNF, NT-3, sTNFR60, sTNFR80 and TNFR2 = 2(3.9%); remaining biomarkers = 1(1.98%). We have also added a summary table with the number of studies included in the systematic review and meta-analysis, separated by neurotrophic factor, mood state and symptoms severity. Please, see Table S2 in the supplementary material.
- Psychometric measures were different among studies, but it is not clear how these were standardized (I.e. severity of mood symptoms)
Response: We thank the reviewer for the comments. Montgomery Asberg Depression Rating Scale (MADRS) scores were converted to Hamilton Depression Rating Scale (HDRS) using the conversion methods developed by Leucht and colleagues in this paper. We have made changes accordingly in the methods section. Please, see line 181.
Leucht, S., Fennema, H., Engel, R. R., Kaspers-Janssen, M., & Szegedi, A. (2018). Translating the HAM-D into the MADRS and vice versa with equipercentile linking. Journal of affective disorders, 226, 326–331. https://doi.org/10.1016/j.jad.2017.09.042
- The discussion is somewhat confusing, the authors state that “This result is consistent with one of the most replicated findings in the literature, the decreased peripheral levels of BDNF in BD”. However, this meta-analysis did not show a difference in depression and mania compared to euthymia.
Response: The reviewer is right that the statement is confusing. We have changed the sentence by: “This result is consistent with one of the most replicated findings in the literature, the decreased peripheral levels of BDNF and its association with depressive symptoms”. Please, see line 451.
Reviewer 3 Report
16 May 2022
Regarding the review of manuscript ‘Clinical value of inflammatory and neurotrophic biomarkers in bipolar disorder: A systematic review and meta-analysis’ by Vega-Núñez A et al., submitted to Biomedicines
Manuscript ID: biomedicines-1714169
Dear Authors,
Vega-Núñe and colleagues in the present review entitled ‘Clinical value of inflammatory and neurotrophic biomarkers in bipolar disorder: A systematic review and meta-analysis’, investigated the current status of knowledge of inflammatory and neurotrophic factors as reliable biomarkers that can characterize clinical features and risk factors for bipolar disorder (BP). For this reason, the authors selected some evidence that explored inflammatory and neurothophic factors in patients with a diagnosis of BD, and selected 18 articles for a meta-analysis. Results showed that the brain-derived neurotrophic factor (BDNF) and cytokines can differentiate BD in three subtypes, showing a negative correlation between circulating BDNF levels and depression severity score and therefore suggesting a key role for BDNF in the depressive component of BD. Controversial results in other inflammatory factors point to the need to analyze additional biomarkers to distinguish subtypes and phases of BD.
The main strength of this manuscript is that it addresses an interesting and timely question, providing a captivating interpretation and describing how circulating inflammatory and neurotrophic factors can be considered reliable biomarkers for clinical parameters (subtypes, phases, severity, episodes, among others) of bipolar disorder. In general, I think the idea of this systematic review is really interesting and the authors’ fascinating observations on this timely topic may be of interest to the readers of Biomedicines. However, some comments, as well as some crucial evidence that should be included to support the author’s argumentation, needed to be addressed to improve the quality of the manuscript, its adequacy, and its readability prior to the publication in the present form, in particular reshaping parts of the ‘Introduction’ and ‘Discussion’ sections by adding more evidence.
Please consider the following comments:
- Abstract: According to the Journal’s guidelines, the abstract should be a total of about 200 words maximum, but the current one includes 253 words. Please, correct it. In addition, I recommend presenting the rationale of this study in the background or the aim and the potential in the conclusion.
- Keywords: I recommend listing up to ten keywords.
- A graphical abstract is highly recommended.
- Introduction: The ‘Introduction’ section is well-written and nicely presented, with a good balance of descriptive text and information about epidemiology and pathogenesis of mood disorders. Even though the authors decided to take a narrow view of mechanisms involved in biological dysregulation in bipolar disorder, I believe that a deeper examination of the mechanisms underlying impairments in emotional learning, memory and how this ability, together with deficient inhibitory control, are core factors in different psychopathologies, such as bipolar disorder, would provide a useful background. Interestingly, results from a recent review outlined typical dysfunctional behaviors, such as deficit in action control and motor inhibition, that are associated with psychopathological and psychiatric conditions, which are characterized by severe impulsivity problems that can determine significant impairment or distress (due to poor regulation and capacity of control, which can be intensified in the presence of emotional stimuli). Accordingly, another recent review (https://doi.org/10.3390/biomedicines10030627), focused on pathological mechanisms underlying altered emotion perception, which is significantly impaired in brain-damaged patients, are related to amygdala and superior temporal sulcus dysfunctions. Finally, I also believe that a recent perspective manuscript on the metabolic pathways involved in the pathogenesis of a wide range of diseases might be of interest. Moreover, if they deem it appropriate, authors can also check additional studies that have focused on searching for biomarkers, developing precision diagnostics, and underlying processes causing impairments in social cognition and social functioning, associated with various psychiatric, neurological and neurodegenerative illnesses (https://doi.org/10.3390/life11121365; https://doi.org/10.3390/biomedicines9070734).
- Search strategy: I suggest Authors to reorganize/rewrite this paragraph because, as it stands, it appears to be too much dispersive and describes the research procedures in an excessively broad way. Also, I would ask them to name the authors who conducted search with initials.
- Results: In my opinion, this section is well organized; however, as the authors have pointed out, only 18 studies were included in the meta-analysis. Therefore, to ensure in-depth understanding and replicability of the findings, I suggest better describing in detail the few studies reported in this review, by providing a detailed description of the hypothesis, the strategies used to study BD, the results and their implications. Furthermore, the authors need to present the risk of bias assessment for the selected study. Also, in my opinion, it is necessary for the authors to present their findings using summary tables.
- Discussion: In my opinion, this systematic review would be more compelling and useful to a broad readership if the authors moved beyond and discussed theoretical and methodological avenues in need of refinement, using this evidence to suggest a path forward. In this regard, I believe that it would have been essential to investigate the neurobiology of mental disorders in humans, and examine the implementation of new therapeutic techniques, such as Non-invasive brain stimulation, that operate to ameliorate the symptoms of mental and neurological disorders. In this regard, I would suggest evidence from recent studies that have examined NIBS efficacy: a recent review described the potential and effectiveness of non-invasive brain stimulation (NIBS) to interfere and modulate the abnormal activity of neural circuits (i.e., amygdala-mPFC-hippocampus) involved in the acquisition and consolidation of fear memories, which are altered in many mood psychiatric disorders (i.e., bipolar disorder, anxiety disorder, specific phobias, post-traumatic stress disorder or depression). Similarly, another recent study illustrated the therapeutic potential of NIBS as a valid alternative in the treatment of abnormally persistent fear memories that characterized those patients with anxiety disorders that do not respond to psychotherapy and/or drug treatments. In addition to the previously mentioned literature, authors can also see these additional studies that have focused on this topic. These findings highlight how NIBS and are a valuable tool in research and has potential diagnostic and therapeutic applications for many mood psychiatry disorders, including bipolar disorder, depression or anxiety.
- I think the ‘Conclusions’ paragraph would benefit from some thoughtful as well as in-depth considerations by the authors, because as it stands, it lists down all the main findings of the research, without really stressing the theoretical significance of the study. Authors should make an effort, trying to explain the theoretical implication as well as the translational application of their research.
- Regarding the Figures and Tables: please provide an explanatory title for each table and figure within the text. Also, please provide higher-quality images because, as it stands, the readers may have difficulty comprehending them.
- The reference list is incorrect: authors should check the Journal’s guidelines again and provide the abbreviated journal name in italics, the year of publication in bold, the volume number in italics.
Overall, the manuscript contains four figures, one table and 72 references. In my opinion, the manuscript might carry important value describing how circulating inflammatory and neurotrophic factors can be considered reliable biomarkers for clinical parameters (subtypes, phases, severity, episodes, among others) of bipolar disorder.
I hope that, after these careful revisions, this paper can meet the Journal’s high standards for publication.
I am available for a new round of revision of this review.
I declare no conflict of interest regarding this manuscript.
Best regards,
Reviewer
Author Response
Vega-Núñez and colleagues in the present review entitled ‘Clinical value of inflammatory and neurotrophic biomarkers in bipolar disorder: A systematic review and meta-analysis’, investigated the current status of knowledge of inflammatory and neurotrophic factors as reliable biomarkers that can characterize clinical features and risk factors for bipolar disorder (BP). For this reason, the authors selected some evidence that explored inflammatory and neurothophic factors in patients with a diagnosis of BD, and selected 18 articles for a meta-analysis. Results showed that the brain-derived neurotrophic factor (BDNF) and cytokines can differentiate BD in three subtypes, showing a negative correlation between circulating BDNF levels and depression severity score and therefore suggesting a key role for BDNF in the depressive component of BD. Controversial results in other inflammatory factors point to the need to analyze additional biomarkers to distinguish subtypes and phases of BD.
The main strength of this manuscript is that it addresses an interesting and timely question, providing a captivating interpretation and describing how circulating inflammatory and neurotrophic factors can be considered reliable biomarkers for clinical parameters (subtypes, phases, severity, episodes, among others) of bipolar disorder. In general, I think the idea of this systematic review is really interesting and the authors’ fascinating observations on this timely topic may be of interest to the readers of Biomedicines. However, some comments, as well as some crucial evidence that should be included to support the author’s argumentation, needed to be addressed to improve the quality of the manuscript, its adequacy, and its readability prior to the publication in the present form, in particular reshaping parts of the ‘Introduction’ and ‘Discussion’ sections by adding more evidence. Please consider the following comments:
1. Abstract: According to the Journal’s guidelines, the abstract should be a total of about 200 words maximum, but the current one includes 253 words. Please, correct it. In addition, I recommend presenting the rationale of this study in the background or the aim and the potential in the conclusion.
Response: Abstract has been updated following Journal’s guidelines.
2. Keywords: I recommend listing up to ten keywords.
Response: We have revised the list of keywords following the recommendation of the reviewer. Each keyword is separated by semicolons.
3. A graphical abstract is highly recommended.
Response: Thank you for the comment. A graphical abstract has been uploaded.
4. Introduction: The ‘Introduction’ section is well-written and nicely presented, with a good balance of descriptive text and information about epidemiology and pathogenesis of mood disorders. Even though the authors decided to take a narrow view of mechanisms involved in biological dysregulation in bipolar disorder, I believe that a deeper examination of the mechanisms underlying impairments in emotional learning, memory and how this ability, together with deficient inhibitory control, are core factors in different psychopathologies, such as bipolar disorder, would provide a useful background. Interestingly, results from a recent review outlined typical dysfunctional behaviors, such as deficit in action control and motor inhibition, that are associated with psychopathological and psychiatric conditions, which are characterized by severe impulsivity problems that can determine significant impairment or distress (due to poor regulation and capacity of control, which can be intensified in the presence of emotional stimuli). Accordingly, another recent review (https://doi.org/10.3390/biomedicines10030627), focused on pathological mechanisms underlying altered emotion perception, which is significantly impaired in brain-damaged patients, are related to amygdala and superior temporal sulcus dysfunctions. Finally, I also believe that a recent perspective manuscript on the metabolic pathways involved in the pathogenesis of a wide range of diseases might be of interest. Moreover, if they deem it appropriate, authors can also check additional studies that have focused on searching for biomarkers, developing precision diagnostics, and underlying processes causing impairments in social cognition and social functioning, associated with various psychiatric, neurological and neurodegenerative illnesses (https://doi.org/10.3390/life11121365; https://doi.org/10.3390/biomedicines9070734).
Response: We thank the reviewer for the comments and the recommended papers. Introduction section has been improved following the reviewer recommendations. Useful information about the mechanisms underlying emotional and cognitive impairments has been incorporated by reviewing the recommended articles. Please, see lines from 56 and 79. Additional references:
23. Lima, I.M.M.; Peckham, A.D.; Johnson, S.L. Cognitive Deficits in Bipolar Disorders: Implications for Emotion. Clin. Psychol. Rev. 2018, 59, 126–136, doi:10.1016/j.cpr.2017.11.006.
24. Baune, B.T.; Malhi, G.S. A Review on the Impact of Cognitive Dysfunction on Social, Occupational, and General Functional Outcomes in Bipolar Disorder. Bipolar Disord. 2015, 17 Suppl 2, 41–55, doi:10.1111/bdi.12341.
25. Depp, C.A.; Mausbach, B.T.; Harmell, A.L.; Savla, G.N.; Bowie, C.R.; Harvey, P.D.; Patterson, T.L. Meta-Analysis of the Association between Cognitive Abilities and Everyday Functioning in Bipolar Disorder. Bipolar Disord. 2012, 14, 217–226, doi:10.1111/j.1399-5618.2012.01011.x.
26. Mackala, S.A.; Torres, I.J.; Kozicky, J.; Michalak, E.E.; Yatham, L.N. Cognitive Performance and Quality of Life Early in the Course of Bipolar Disorder. J. Affect. Disord. 2014, 168, 119–124, doi:10.1016/j.jad.2014.06.045.
27. Mora, E.; Portella, M.J.; Forcada, I.; Vieta, E.; Mur, M. Persistence of Cognitive Impairment and Its Negative Impact on Psychosocial Functioning in Lithium-Treated, Euthymic Bipolar Patients: A 6-Year Follow-up Study. Psychol. Med. 2013, 43, 1187–1196, doi:10.1017/S0033291712001948.
28. Johnson, S.L.; Carver, C.S. Emotion-Relevant Impulsivity Predicts Sustained Anger and Aggression after Remission in Bipolar I Disorder. J. Affect. Disord. 2016, 189, 169–175, doi:10.1016/j.jad.2015.07.050.
29. Johnson, S.L.; Carver, C.S.; Tharp, J.A. Suicidality in Bipolar Disorder: The Role of Emotion-Triggered Impulsivity. Suicide Life. Threat. Behav. 2017, 47, 177–192, doi:10.1111/sltb.12274.
30. Miller, J.N.; Black, D.W. Bipolar Disorder and Suicide: A Review. Curr. Psychiatry Rep. 2020, 22, 6, doi:10.1007/s11920-020-1130-0.
31. Ramírez-Martín, A.; Ramos-Martín, J.; Mayoral-Cleries, F.; Moreno-Küstner, B.; Guzman-Parra, J. Impulsivity, Decision-Making and Risk-Taking Behaviour in Bipolar Disorder: A Systematic Review and Meta-Analysis. Psychol. Med. 2020, 50, 2141–2153, doi:10.1017/S0033291720003086.
32. Zakowicz, P.; SkibiÅ„ska, M.; Wasicka-Przewoźna, K.; Skulimowski, B.; WaÅ›niewski, F.; Chorzepa, A.; RóżaÅ„ski, M.; Twarowska-Hauser, J.; Pawlak, J. Impulsivity as a Risk Factor for Suicide in Bipolar Disorder. Front. Psychiatry 2021, 12, 706933, doi:10.3389/fpsyt.2021.706933.
33. Gvion, Y.; Levi-Belz, Y.; Hadlaczky, G.; Apter, A. On the Role of Impulsivity and Decision-Making in Suicidal Behavior. World J. Psychiatry 2015, 5, 255–259, doi:10.5498/wjp.v5.i3.255.
34. Battaglia, S.; Fabius, J.H.; Moravkova, K.; Fracasso, A.; Borgomaneri, S. The Neurobiological Correlates of Gaze Perception in Healthy Individuals and Neurologic Patients. Biomedicines 2022, 10, 627, doi:10.3390/biomedicines10030627.
35. Tanaka, M.; Tóth, F.; Polyák, H.; Szabó, Á.; Mándi, Y.; Vécsei, L. Immune Influencers in Action: Metabolites and Enzymes of the Tryptophan-Kynurenine Metabolic Pathway. Biomedicines 2021, 9, 734, doi:10.3390/biomedicines9070734.
43. Bora, E. Developmental Trajectory of Cognitive Impairment in Bipolar Disorder: Comparison with Schizophrenia. Eur. Neuropsychopharmacol. J. Eur. Coll. Neuropsychopharmacol. 2015, 25, 158–168, doi:10.1016/j.euroneuro.2014.09.007.
44. Joseph, M.F.; Frazier, T.W.; Youngstrom, E.A.; Soares, J.C. A Quantitative and Qualitative Review of Neurocognitive Performance in Pediatric Bipolar Disorder. J. Child Adolesc. Psychopharmacol. 2008, 18, 595–605, doi:10.1089/cap.2008.064.
45. Post, R.M.; Fleming, J.; Kapczinski, F. Neurobiological Correlates of Illness Progression in the Recurrent Affective Disorders. J. Psychiatr. Res. 2012, 46, 561–573, doi:10.1016/j.jpsychires.2012.02.004.
46. Arts, B.; Jabben, N.; Krabbendam, L.; van Os, J. Meta-Analyses of Cognitive Functioning in Euthymic Bipolar Patients and Their First-Degree Relatives. Psychol. Med. 2008, 38, 771–785, doi:10.1017/S0033291707001675.
47. Balanzá-Martínez, V.; Rubio, C.; Selva-Vera, G.; Martinez-Aran, A.; Sánchez-Moreno, J.; Salazar-Fraile, J.; Vieta, E.; Tabarés-Seisdedos, R. Neurocognitive Endophenotypes (Endophenocognitypes) from Studies of Relatives of Bipolar Disorder Subjects: A Systematic Review. Neurosci. Biobehav. Rev. 2008, 32, 1426–1438, doi:10.1016/j.neubiorev.2008.05.019.
48. Bora, E.; Yucel, M.; Pantelis, C. Cognitive Endophenotypes of Bipolar Disorder: A Meta-Analysis of Neuropsychological Deficits in Euthymic Patients and Their First-Degree Relatives. J. Affect. Disord. 2009, 113, 1–20, doi:10.1016/j.jad.2008.06.009.
5. Search strategy: I suggest Authors to reorganize/rewrite this paragraph because, as it stands, it appears to be too much dispersive and describes the research procedures in an excessively broad way. Also, I would ask them to name the authors who conducted search with initials.
Response: Thank you for the suggestion. We have included the Author´s initials and moved the search strategy to the supplementary material.
6. Results: In my opinion, this section is well organized; however, as the authors have pointed out, only 18 studies were included in the meta-analysis. Therefore, to ensure in-depth understanding and replicability of the findings, I suggest better describing in detail the few studies reported in this review, by providing a detailed description of the hypothesis, the strategies used to study BD, the results and their implications. Furthermore, the authors need to present the risk of bias assessment for the selected study. Also, in my opinion, it is necessary for the authors to present their findings using summary tables.
Response: The revised manuscript has been revised following the reviewer’s comments. A summary of a detailed description of the studies selected for each meta-analysis was reported (Please, see lines 263, 287, 295, 326 and 337). A detailed description of the 18 studies (2, 11, 12, 13, 18, 21, 22, 26, 30b, 32, 33a, 34, 35, 40, 42, 43, 44a, 45) can be also found in Table 1. An additional summary table showing the number of studies included in the systematic review, separated by neurotrophic factor, mood state and symptoms severity. Please, see Table S2 in the supplementary material. Finally, we have included a traffic light plot (Figure S1) and a summary plot (Figure S2) to assess risk of bias following the Risk of Bias in Non-randomised Studies (ROBINS-I) tool. Please, see supplementary material.
7. Discussion: In my opinion, this systematic review would be more compelling and useful to a broad readership if the authors moved beyond and discussed theoretical and methodological avenues in need of refinement, using this evidence to suggest a path forward. In this regard, I believe that it would have been essential to investigate the neurobiology of mental disorders in humans, and examine the implementation of new therapeutic techniques, such as Non-invasive brain stimulation, that operate to ameliorate the symptoms of mental and neurological disorders. In this regard, I would suggest evidence from recent studies that have examined NIBS efficacy: a recent review described the potential and effectiveness of non-invasive brain stimulation (NIBS) to interfere and modulate the abnormal activity of neural circuits (i.e., amygdala-mPFC-hippocampus) involved in the acquisition and consolidation of fear memories, which are altered in many mood psychiatric disorders (i.e., bipolar disorder, anxiety disorder, specific phobias, post-traumatic stress disorder or depression). Similarly, another recent study illustrated the therapeutic potential of NIBS as a valid alternative in the treatment of abnormally persistent fear memories that characterized those patients with anxiety disorders that do not respond to psychotherapy and/or drug treatments. In addition to the previously mentioned literature, authors can also see these additional studies that have focused on this topic. These findings highlight how NIBS and are a valuable tool in research and has potential diagnostic and therapeutic applications for many mood psychiatry disorders, including bipolar disorder, depression or anxiety.
Response: Thank you very much for your thoughtful comment. Unfortunately, we think the review is very wide and it is out of our objective to discuss treatment options and neurobiology of BD in deep. We have added in the discussion a sentence that we think summarize the idea of the reviewer: “In general, more studies are necessary to understand precisely the neurobiology of BD and their interactions with stress and clinical factors. This improvement will presumably translate to more sophisticated treatments.” Please, see line 500.
8. I think the ‘Conclusions’ paragraph would benefit from some thoughtful as well as in-depth considerations by the authors, because as it stands, it lists down all the main findings of the research, without really stressing the theoretical significance of the study. Authors should make an effort, trying to explain the theoretical implication as well as the translational application of their research.
Response: Thank you for the suggestion. We have modified the conclusions and we have considered the implications of the results in depth. Line 547.
9. Regarding the Figures and Tables: please provide an explanatory title for each table and figure within the text. Also, please provide higher-quality images because, as it stands, the readers may have difficulty comprehending them.
Response: We have included the title of each Table and Figure. The quality of the images has been updated.
10. The reference list is incorrect: authors should check the Journal’s guidelines again and provide the abbreviated journal name in italics, the year of publication in bold, the volume number in italics.
Response: The reviewer is right. We have checked the Journal’s guidelines and modified the references accordingly.
11. Overall, the manuscript contains four figures, one table and 72 references. In my opinion, the manuscript might carry important value describing how circulating inflammatory and neurotrophic factors can be considered reliable biomarkers for clinical parameters (subtypes, phases, severity, episodes, among others) of bipolar disorder.
Response: We thank the reviewer for the helpful comments.
Reviewer 4 Report
Vega-Núñez et al. systematically reviewed the use of inflammatory and neurotrophic biomarkers in bipolar disorder and argue that BDNF levels should be considered a key biomarker of depression in bipolar disorder. Unfortunately, much of the literature reviewed with respect to other inflammatory markers is quite mixed, making firm conclusions in this regard very difficult. Specifically, many articles reviewed here had the following weaknesses:
- the correlation between bipolar disorder and serum markers is based on various end-point measures, including in certain cases non-relevant parameters (such as BMI, daily caloric intake) or clinical interviews which somewhat hampers the reliability of the results.
- some of the studies pursued the correlations of interests in very different populations regarding the age or the aims of the studies were very different - assessing cognitive function, the severity of symptoms, etc.
At the moment, it does not appear that any firm conclusions can be drawn about the role of BDNF in BD. Eventually, it can be used as a biomarker.
Nonetheless, it should be emphasized in Conclusions and also in the Abstract that much of the evidence reviewed is mixed and contradictory, the possible reasons for these differences, and future studies need to be designed to resolve these inconsistencies (e.g., account for the major confounds not controlled for in past studies). Especially, the Abstract should be rewritten, because in this form, it suggests that interleukins, TNFa, and even sVCAM-1 are reliable biomarkers of the disease.
Author Response
Vega-Núñez et al. systematically reviewed the use of inflammatory and neurotrophic biomarkers in bipolar disorder and argue that BDNF levels should be considered a key biomarker of depression in bipolar disorder. Unfortunately, much of the literature reviewed with respect to other inflammatory markers is quite mixed, making firm conclusions in this regard very difficult. Specifically, many articles reviewed here had the following weaknesses:
- The correlation between bipolar disorder and serum markers is based on various end-point measures, including in certain cases non-relevant parameters (such as BMI, daily caloric intake) or clinical interviews which somewhat hampers the reliability of the results.
Response: Thank you for the suggestions. These and other potential confounders (including underpowered sample sizes, inadequate testing correction, inconclusive results compared to other psychiatric disorders and different populations with respect to age, sex, metabolic alterations, cognitive function and symptom severity that are not accounted for by BD studies) may explain the lack of consistent findings and the difficulties in establishing reliable biomarkers for BD. Some of these weaknesses pointed out are included in the limitations section (line 541). We have added that “Some clinical variables are assessed only based in clinical interviews which can compromise the reliability and validity of the results”. To further addresses these limitations authors have performed the risk of bias assessment for the selected studies. We have included a traffic light plot (Figure S1) and a summary plot (Figure S2) to assess risk of bias following the Risk of Bias in Non-randomised Studies (ROBINS-I) tool. Please, see supplementary material.
- Some of the studies pursued the correlations of interests in very different populations regarding the age or the aims of the studies were very different - assessing cognitive function, the severity of symptoms, etc.
Response: The reviewer is right. Following the previous response, we have included in the limitation section the sentence: “Many studies included small samples and aims between them were different”. Line 531.
- At the moment, it does not appear that any firm conclusions can be drawn about the role of BDNF in BD. Eventually, it can be used as a biomarker.
Response: Thank you for the suggestion. The clinical value of BDNF and other inflammatory biomarkers has yet to be demonstrated by highly controlled studies that follow a precise phenotyping (type, phase, severity, and comorbidities) of BD patients. In the revised manuscript, we have included an additional summary table (Table S2) showing the number of studies included in the systematic review, separated by neurotrophic factor, mood state and symptoms severity. A highlighted conclusion of the present revision is that the known evidence for BDNF down-regulation in major depression remains blurred or inconsistent in BD patients. As noted above, this inconsistency is based on a number of important uncontrolled confounders. However, despite the confounding factors, the main finding of this systematic review and meta-analysis is the negative correlation found between BDNF levels and severity of depressive symptoms in BD patients. As indicated by the reviewer, this evidence suggests that BDNF may be an eventual biomarker for BD. We have included in the conclusions that “more studies need to be conducted to stablish the clinical value of this biomarker (BDNF) among others”.
- Nonetheless, it should be emphasized in Conclusions and also in the Abstract that much of the evidence reviewed is mixed and contradictory, the possible reasons for these differences, and future studies need to be designed to resolve these inconsistencies (e.g., account for the major confounds not controlled for in past studies). Especially, the Abstract should be rewritten, because in this form, it suggests that interleukins, TNFa, and even sVCAM-1 are reliable biomarkers of the disease.
Response: We have rewritten the abstract following reviewer’s comments. We have also changed the conclusions including the suggestion of the reviewer.
Round 2
Reviewer 2 Report
No further comments
Author Response
No further comments
Response: Thank you for your revision.
Reviewer 3 Report
29 May 2022
Regarding the review of manuscript ‘Clinical value of inflammatory and neurotrophic biomarkers in bipolar disorder: A systematic review and meta-analysis’ by Vega-Núñez A et al., submitted to Biomedicines
Manuscript ID: biomedicines-1714169
Dear Authors,
I am very pleased to see that the Authors have welcomed my suggestions and have clarified most of the issues I raised in my first round of this review. I believe that this systematic review entitled ‘Clinical value of inflammatory and neurotrophic biomarkers in 3 bipolar disorder: A systematic review and meta-analysis’ does an excellent work describing how circulating inflammatory and neurotrophic factors can be considered reliable biomarkers for clinical parameters (subtypes, phases, severity, episodes, among others) of bipolar disorder.
I only have one last minor suggestion to further improve the theoretical background about biological dysregulation in bipolar disorder: for this reason, I suggest again to add some findings from additional evidence that have examined the biomarkers and the mechanisms underlying cognitive impairments in different psychopathologies, such as bipolar disorder (https://doi.org/10.3390/ijms23105460; https://doi.org/10.1016/j.brat.2021.103963; https://doi.org/10.3390/biomedicines9050517).
Overall, this is a timely and needed study, and I look forward to seeing further studies on this issue by these authors in the future.
I am always available for other revisions of such as interesting and important reviews.
Thank You for your work.
I declare no conflict of interest regarding this manuscript.
Best regards,
Reviewer
Author Response
I only have one last minor suggestion to further improve the theoretical background about biological dysregulation in bipolar disorder: for this reason, I suggest again to add some findings from additional evidence that have examined the biomarkers and the mechanisms underlying cognitive impairments in different psychopathologies, such as bipolar disorder (https://doi.org/10.3390/ijms23105460; https://doi.org/10.1016/j.brat.2021.103963; https://doi.org/10.3390/biomedicines9050517).
Response: We thank the reviewer for the very interesting and timely studies. The revised manuscript has been updated accordingly. Please, see lines 58, 73 and 93.
Overall, this is a timely and needed study, and I look forward to seeing further studies on this issue by these authors in the future.
Response: Thank you for your kind words. We are working in that direction.